# Epilepsy and intellectual disability linked protein Shrm4 interaction with GABA$_B$Rs shapes inhibitory neurotransmission

Jonathan Zapata[1,2,*], Edoardo Moretto[1,2,*], Saad Hannan[3], Luca Murru[1,2], Anna Longatti[1,4], Davide Mazza[5], Lorena Benedetti[1,2], Matteo Fossati[1,2], Christopher Heise[1,2], Luisa Ponzoni[6], Pamela Valnegri[1,2], Daniela Braida[2], Mariaelvina Sala[1,2], Maura Francolini[1,2], Jeffrey Hildebrand[7], Vera Kalscheuer[8], Francesca Fanelli[9], Carlo Sala[1,2], Bernhard Bettler[10], Silvia Bassani[1,2], Trevor G. Smart[3] & Maria Passafaro[1,2]

Shrm4, a protein expressed only in polarized tissues, is encoded by the *KIAA1202* gene, whose mutations have been linked to epilepsy and intellectual disability. However, a physiological role for Shrm4 in the brain is yet to be established. Here, we report that Shrm4 is localized to synapses where it regulates dendritic spine morphology and interacts with the C terminus of GABA$_B$ receptors (GABA$_B$Rs) to control their cell surface expression and intracellular trafficking via a dynein-dependent mechanism. Knockdown of Shrm4 in rat severely impairs GABA$_B$R activity causing increased anxiety-like behaviour and susceptibility to seizures. Moreover, Shrm4 influences hippocampal excitability by modulating tonic inhibition in dentate gyrus granule cells, in a process involving crosstalk between GABA$_B$Rs and extrasynaptic δ-subunit-containing GABA$_A$Rs. Our data highlights a role for Shrm4 in synaptogenesis and in maintaining GABA$_B$R-mediated inhibition, perturbation of which may be responsible for the involvement of Shrm4 in cognitive disorders and epilepsy.

[1] CNR, Institute of Neuroscience, Via Vanvitelli, 32, 20129 Milano, Italy. [2] Department of Medical Biotechnology and Translational Medicine (BIOMETRA), Università di Milano, Via Vanvitelli, 32, 20129 Milano, Italy. [3] Department of Neuroscience, Physiology and Pharmacology, University College London, Gower Street, London WC1E 6BT, UK. [4] Department of Pharmacological and Biomolecular Sciences (DiSFeB)—Università di Milano, Via Balzaretti 9, 20133 Milano, Italy. [5] Centro di Imaging Sperimentale e Università Vita-Salute San Raffaele, Istituto Scientifico Ospedale San Raffaele, Via Olgettina 60, Milano 20132, Italy. [6] Fondazione Umberto Veronesi, Piazza Velasca 5, Milan 20122, Italy. [7] Department of Biological Sciences, University of Pittsburgh, 103A Life Sciences Annex 4249, Fifth Avenue Pittsburgh, Pennsylvania 15260, USA. [8] Department of Human Molecular Genetics, Max Planck Institute for Molecular Genetics, Ihnestrasse 63-73, D-14195 Berlin, Germany. [9] Department of Life Sciences, University of Modena and Reggio Emilia, via Campi 103, 41125 Modena, Italy. [10] Department of Biomedicine, University of Basel, Klingelbergstrasse 50/70, CH-4056 Basel, Switzerland. * These authors contributed equally to this work. Correspondence and requests for materials should be addressed to M.P. (email: m.passafaro@in.cnr.it).

The actin-binding proteins Shroom (Shrm) play an important role in cytoskeletal organization and consist of an N-terminal PDZ domain, a central Apx/Shrm Domain 1 (ASD1) and a C-terminal ASD2 domain[1]. There are four evolutionarily conserved Shrm proteins (Shrm1-4)[2] that are localized to polarized tissues, including neurons[1,3]. Of these, only Shrm4 lacks the actin-targeting ASD1 motif, and the role of its PDZ domain is unknown. Murine Shrm4 possesses putative binding sites for EVH1 (poly-proline rich domain), a PDZ (SNF) binding motif[4] and a stretch of glutamine and glutamate residues preceding the C-terminal ASD2 motif that is unique to Shrm4, but not other family members[4]. Shrm4 is ubiquitously expressed throughout embryonic and adult murine brains[2] and also binds to F-actin in non-neuronal cells[4].

The importance of Shrm4 is illustrated by two *de novo* balanced X-chromosomal translocations, which disrupts the *KIAA1202* gene (Xp11.2) that encodes for Shrm4. In addition, a pathogenic missense mutation was identified in an unrelated large family, with carriers exhibiting mild-to-severe intellectual disability (ID) and increased susceptibility to seizures[2,5]. Recent studies reinforce the role of Shrm4 in ID[6–8], however, how disruption of *KIAA1202* causes these neuropathological conditions is unknown. Indeed, the role of Shrm4 in the brain is also unknown, but given the pathological profile, it may regulate GABA-mediated inhibition[9,10]. GABA activates ionotropic $GABA_A$ (ref. 11) ($GABA_A R$) and metabotropic $GABA_B$ receptors ($GABA_B Rs$)[12] to control inhibition which is important for synaptic plasticity[13,14]. The importance of these receptors is emphasized during dysfunction, which occurs in different neurological disease[14,15].

Here, we report that Shrm4 interacts with $GABA_B Rs$ to facilitate trafficking to dendrites using a dynein-dependent mechanism. For cell surface expression, $GABA_B Rs$ are obligate heterodimers comprising $GABA_B R1$ ($GABA_{B1}$) and R2 ($GABA_{B2}$) subunits that mediate long-lasting synaptic inhibition[16]. However, the motor-protein-dependent trafficking of these receptors is not fully understood.

We found that loss of Shrm4 compromises $GABA_B R$ delivery to postsynaptic compartments, impairs $GABA_B R$-mediated $K^+$ currents and $GABA_A R$-mediated tonic inhibition in the hippocampus, and reduces dendritic spine density altering the composition of synaptic proteins resulting in increased anxiety-like behaviour and susceptibility to seizures in rats. Our study suggests a possible underlying mechanism by which Shrm4 may be involved in epilepsy and ID.

## Results

**Shrm4 is an interacting partner of $GABA_B$ receptors**. We first investigated the subcellular localization of Shrm4 in cultured rat hippocampal neurons at 18 days *in vitro* (*DIV*) by immunostaining and found colocalization with presynaptic (Bassoon) and postsynaptic markers of excitatory (PSD-95) and inhibitory ($GABA_A\beta3$) synapses (Supplementary Fig. 1a). By using electron microscopy and post-embedding immunogold methods with an anti-Shrm4 antibody[2], gold nanoparticles were identified in pre- and post-synaptic areas and along dendrites (Supplementary Fig. 1b; Supplementary Table 1). Biochemical fractionation of adult rat brain hippocampi and cortices revealed enrichment of Shrm4 in the postsynaptic density (PSD) fraction further confirming its presence at synapses (Supplementary Fig. 1c).

To explore the role(s) of Shrm4 in neurons, we searched for binding partners using yeast two-hybrid (Y2H) screening. The PDZ domain of Shrm4 (residues 1–91; Fig. 1a) was selected as the bait to screen against an adult human brain cDNA library, as this domain participates in protein–protein interactions. Twenty

positive cDNA clones were isolated; six of these encoded a 100 amino acids stretch of the C-terminal tail of $GABA_{B1}$ present in both splice variants B1a and B1b, which differ by the inclusion of two Sushi domains in the N terminus only in $GABA_{B1a}$ (refs 12,14) (Fig. 1a). The interaction between Shrm4 and the $GABA_{B1}$ C-tail was crucially reliant on the Shrm4-PDZ domain (PDZ domain 1–91: +++; ΔPDZ 91–1,492: negative; ΔASD2 1–1,213: +++; Fig. 1a). We confirmed this interaction using rat brain lysates and found that endogenous Shrm4 co-precipitates with $GABA_{B1a/b}$ (Fig. 1b). $GABA_B R$ antibody specificity was confirmed by western blots from $GABA_{B1a}$ and $GABA_{B1b}$ knockout mice[17] (Supplementary Fig. 1d). Moreover, direct stochastic optical reconstruction microscopy (dSTORM) of 14*DIV* hippocampal neurons revealed that Shrm4 and $GABA_B Rs$ co-cluster in neurons (Fig. 1c; Supplementary Fig. 1e)[18].

GST pull-downs from cultured hippocampal neurons and brain extracts confirmed that Shrm4-PDZ domain alone is sufficient to precipitate $GABA_{B1a/b}$, unlike a mutated Shrm4-PDZ domain (K26G27/AA) (Fig. 1d, panels 1–3). To ensure $GABA_{B1}$ specificity for this interaction, we verified that HA-tagged Shrm4 did not co-precipitate Flag-tagged $GABA_{B2}$ from HEK293 cells when transfected alone (Fig. 1e).

We used immunoprecipitation to determine the minimal region of the $GABA_{B1}$ C-tail that interacts with Shrm4 in HEK293 cells expressing Shrm4-GFP, $GABA_{B2}$ and $GABA_{B1a}$ cDNAs with differing C termini (Fig. 1f). This revealed that $R^{859}LITRGEWQSEA^{870}$ in the C-tail of $GABA_{B1}$ interacted with the Shrm4-PDZ domain (Fig. 1f). To confirm this interaction, we produced a cell-permeable peptide fragment (Tat-peptide) encompassing the Shrm4-$GABA_{B1}$ binding site on GABAB1 (859–870). We verified the efficiency of this peptide in pull-down experiments with GST-PDZ incubating different concentration of Tat control or Tat-859–870 with lysates of HEK293 cells overexpressing $GABA_{B1a}$-GFP and $GABA_{B2}$ (Supplementary Fig. 1f) and a dose-dependence was observed. We then incubated lysates from HEK293 cells overexpressing $GABA_{B1a}$ and $GABA_{B2}$ with this peptide before GST pull-down using Shrm4-PDZ or its mutant form (K26G27/AA). Consistent with previous results, co-precipitation of $GABA_{B1}$ was reduced after preincubation with the Tat-peptide compared with its scrambled control (Fig. 1g). These results reveal a new association between the PDZ domain of Shrm4 and residues 859–870 of the $GABA_{B1}$ C terminus.

**Shrm4 regulates the surface expression of $GABA_B Rs$**. Given that the C-tail of $GABA_{B1}$ is important for trafficking to the cell surface[19,20], we next examined whether Shrm4 regulates $GABA_B R$ trafficking. We designed two shRNAs that specifically targeted rat (shRNA#1 and #2) and human (shRNA#2) *Shrm4* transcripts. To test their effectiveness as Shrm4 silencers, HEK293 cells were transfected with human HA-Shrm4 cDNA (Rescue) with or without each shRNA. Both shRNAs reduced Shrm4 expression while co-transfection of a rescue construct with shRNA#1 restored expression (Supplementary Fig. 2a,b). We extended this approach to neurons, transfecting Scrambled, shRNA#1, shRNA#2 or Rescue constructs at 8*DIV* for later analysis of surface levels of $GABA_{B1}$ at 18*DIV*. Interestingly, both shRNAs reduced surface levels of $GABA_{B1}$ that can be rescued by the re-expression of Shrm4 (Fig. 1h, full neurons in Supplementary Fig. 2c). As an additional control, we tested the effect of knocking down Shrm3 and found no effect on $GABA_{B1}$ surface expression indicating a unique role played by Shrm4 in $GABA_{B1}$ trafficking (Supplementary Fig. 2d). We next used lentiviral delivery to corroborate our immunostaining results. For this we used cell surface biotinylation assays for $GABA_B Rs$ (Fig. 1i). Shrm4 knockdown reduced surface expression of $GABA_B Rs$ without

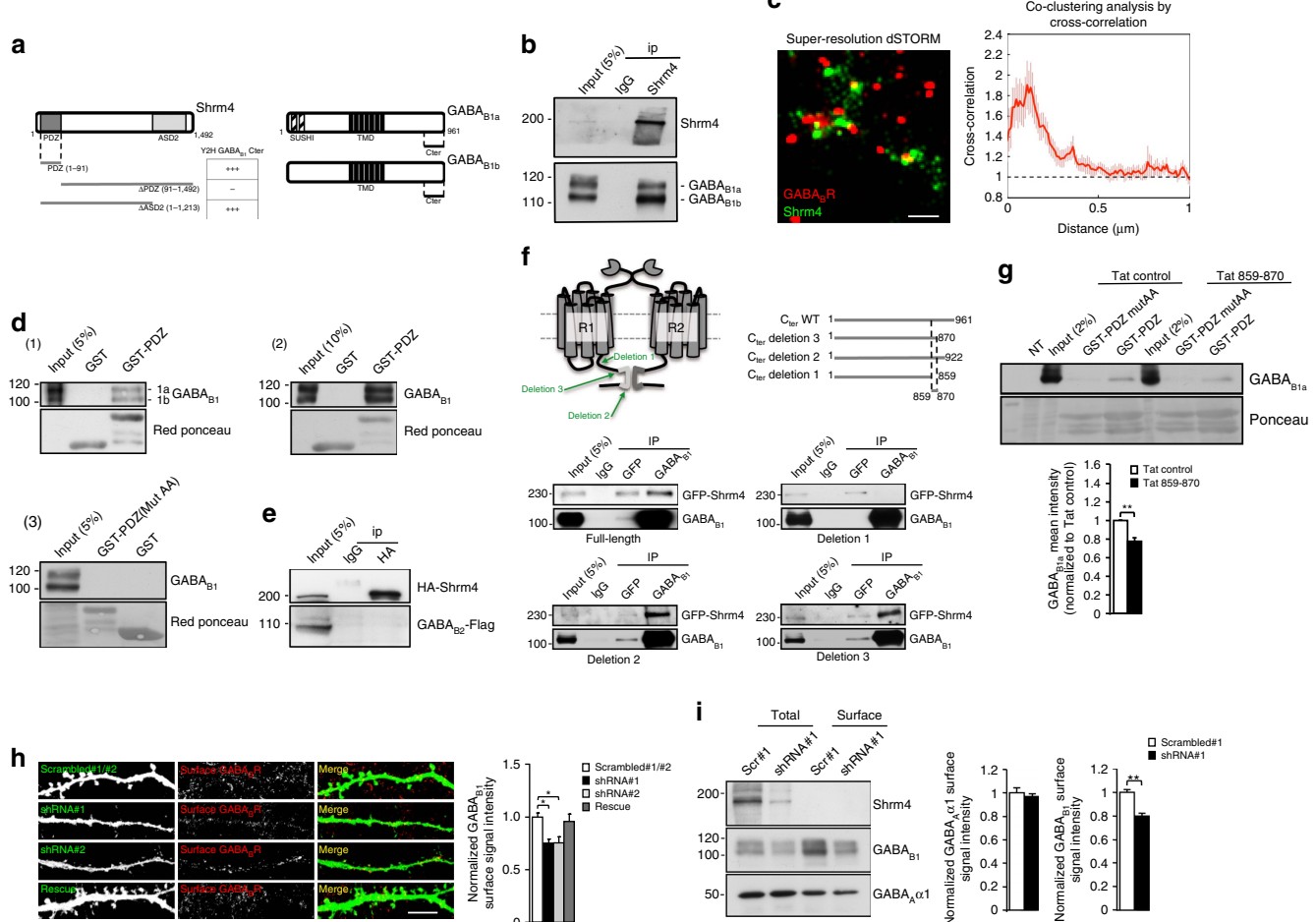

**Figure 1 | Shrm4 interaction with GABA$_B$Rs modulates cell surface expression.** (**a**) Representations of Shrm4 and GABA$_B$R subunit 1a and 1b interacting domains. The PDZ domain of Shrm4 (1–91 aa) was used as bait for Y2H screening on adult human brain cDNA library. Twenty positive clones were isolated; six encoded for a 100 aa stretch of the GABA$_B$R subunit 1 C-terminal tail (both isoforms (1a and 1b)). The PDZ domain and ΔASD2 constructs interact with the C-terminal tail of GABA$_{B1}$ ( + + + ) while ΔPDZ truncated construct does not ( − ). TMD: Transmembrane domains. (**b**) Co-immunoprecipitation experiments on adult rat brain extracts using anti-Shrm4 antibody show Shrm4 and GABA$_{B1}$ association (full blot in Supplementary Fig. 11). (**c**) (Left) dSTORM imaging of GABA$_B$R (red) and Shrm4 (green) on 14DIV rat cultured hippocampal neurons. (Right) Shrm4 and GABA$_B$R puncta co-cluster as evidenced by cross-correlation analysis. Error-bars are s.e.m.; number of regions = 29, number of fields = 3. Scale bar, 0.4 μm. (**d**) GST pull-down experiments on 18DIV rat cultured hippocampal neurons (1) and rat brain (2) extract show that Shrm4-PDZ domain (GST-PDZ; aas 1–91) pulls down both GABA$_{B1}$ isoforms whereas (3) mutant GST-PDZ (AA) does not (full blot in Supplementary Fig. 11). (**e**) Co-immunoprecipitation experiments on HEK293 cells expressing HA-Shrm4 and GABA$_{B2}$-Flag, showing that Shrm4 does not associate with GABA$_{B2}$ in absence of GABA$_{B1}$ (full blot in Supplementary Fig. 11). (**f**) Scheme of truncated GABA$_B$R constructs: deletion 1: Δ859–961; deletion 2: Δ922–961; and deletion 3: Δ870–961 with co-immunoprecipitation results from HEK293 cells expressing Shrm4-GFP and each of the GABA$_{B1a}$ constructs. GFP-Shrm4 immunoprecipitates full-length GABA$_{B1a}$, deletion 2, and deletion 3, but not deletion 1 (full blot in Supplementary Fig. 11). (**g**) (Top) GST pull-down experiments using GST-PDZ or mutant GST-PDZ (AA) on lysates of HEK293 cells overexpressing GABA$_{B1}$ and GABA$_{B2}$ incubated with Tat-control peptide or Tat-859–870 peptide corresponding to the previously identified minimal GABA$_{B1}$-Shrm4 binding region. (Bottom) Histograms showing GABA$_{B1a}$ mean intensity normalized to Tat-control peptide ± s.e.m. n = 3; **P = 0.046; t-test (full blot in Supplementary Fig. 11). (**h**) Surface immunostaining for GABA$_B$R in rat cultured hippocampal neurons at 18DIV transfected with scrambled, shRNA#1, shRNA#2 or rescue constructs at 8DIV. Scale bar, 15 μm. Histograms show mean ± s.e.m. n = 5–15, Scrambled versus shRNA#1 *P = 0.0153; Scrambled versus shRNA#2 *P = 0.0153; One-way ANOVA-Mann–Whitney. (**i**) GABA$_B$R cell surface biotinylation from 18DIV rat cultured hippocampal neurons infected with scrambled#1 (Scr#1) or shRNA#1 (shRNA#1) at 8DIV and western blot. Shrm4 silencing reduces GABA$_{B1}$ surface expression, but does not change total signal for GABA$_{B1}$ or surface (and total) signal for GABA$_A$R α1 subunit. Histograms show mean ± s.e.m.; n = 4; **P = 0.0069; t-test (full blot in Supplementary Fig. 11).

affecting the total expression of GABA$_B$Rs or the total and surface expression of GABA$_A$R α1 subunits used as negative controls (Fig. 1i). Thus, Shrm4 is important for GABA$_{B1}$ trafficking in neurons.

**Shrm4 regulates synaptic structure and protein composition.** Since Shrm4 colocalizes with PSD-95 and Bassoon, we investi-

gated if its knockdown affected excitatory synapses in rat pyramidal hippocampal neurons. In these excitatory neurons, expressing scrambled or shRNA#1 before synaptogenesis (8DIV) did not alter the branching or dendritic diameters at 18DIV (Fig. 2a). We therefore refined our analysis and studied dendritic spines, where defects are commonly linked to ID[21]. First, we investigated the expression levels of PSD-95, AMPA receptor subunit GluA2 and presynaptic markers Bassoon and Synapsin,

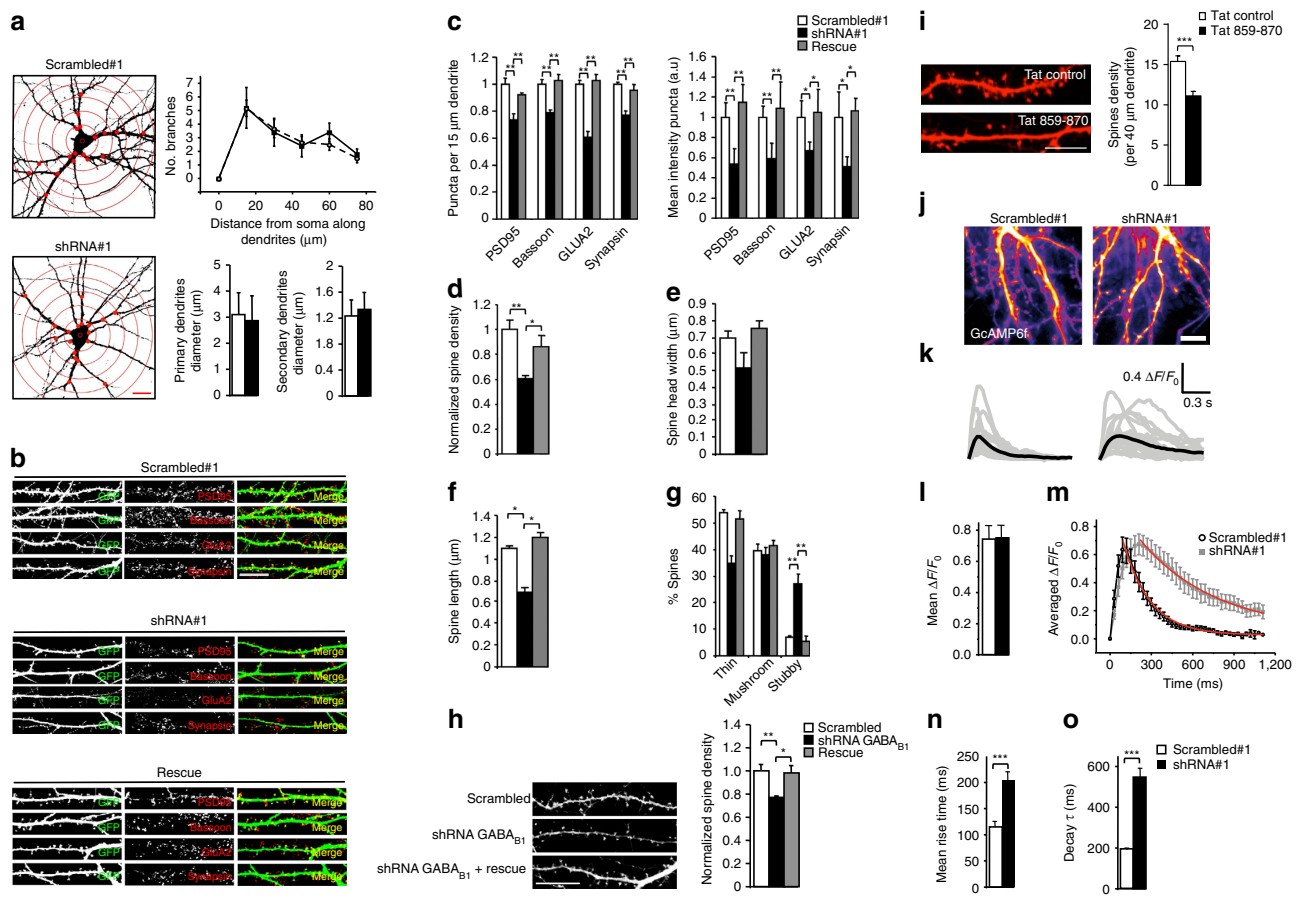

**Figure 2 | Shrm4 is key mediator of dendritic spine formation and morphology.** (**a**) Sholl analyses were performed on 18*DIV* rat hippocampal neurons transfected at 8*DIV* with GFP-coexpressing knockdown shRNA#1 or scrambled#1. No difference in dendritic arborization and diameter was observed between the two conditions. Scale bars, 10 μm. (**b**) Images of rat hippocampal neurons transfected at 8*DIV* with Shrm4 scrambled#1, shRNA#1 or rescue constructs and immunostained for PSD-95, Bassoon, GluA2 and Synapsin post-synaptogenesis at 18*DIV*. Scale bar, 10 μm. (**c**) Numbers of fluorescence puncta/15 μm dendrite (left panel) and mean fluorescence intensity of synaptic markers (right panel). The effect of Shrm4 silencing could be rescued by coexpressing wild-type Shrm4 cDNA resistant to knockdown by shRNA#1 (Data normalized to scrambled#1 controls) Histograms show mean ± s.e.m.; see Supplementary Table 2 for statistical tests. (**d**) Normalized spine density per 20 μm of hippocampal neurons transfected at 8*DIV* for Shrm4 knockdown and rescue. See Supplementary Table 2 for statistical tests. (**e**) Spine head width and (**f**) length for scrambled, knockdown and rescue cells. See Supplementary Table 2 for statistical tests. (**g**) Percentages of mushroom, stubby and thin spines for scrambled, knockdown and rescue cells. Histograms show mean ± s.e.m.; see Supplementary Table 2 for statistical tests. (**h**) (Left) Images of rat hippocampal neurons transfected at 8*DIV* with scrambled, $GABA_{B1}$ knockdown shRNA or rescue constructs and analysed at 18*DIV*. (Right) Histogram showing normalized spine density per 20 μm of hippocampal neurons transfected at 8*DIV* for scrambled, $GABA_{B1}$ knockdown and rescue; Histograms show mean ± s.e.m.; n = 6–8 cells; Scrambled versus shRNA **$P = 0.0056$; shRNA versus rescue *$P = 0.0108$, one-way ANOVA-Mann–Whitney. Scale bars, 10 μm. (**i**) (Left) Images of 17*DIV* rat hippocampal neurons transfected at 5*DIV* with pSuper and treated from 8*DIV* to 13*DIV* with a Tat control or Tat-859–870 peptide. (Right) Histogram showing spine density per 40 μm dendrite of 17*DIV*-treated hippocampal neurons; n = 10; ***$P = 0.0001$; t-test. Scale bar, 10 μm. (**j**) Representative $Ca^{2+}$ signals from hippocampal neurons expressing dsRed-coexpressing knockdown shRNA#1 or scrambled#1 along with GCaMP6 fast under basal conditions in modified Krebs solution (in mM: 140 NaCl, 2.5 CaCl₂, 4.7 KCl, 11 glucose, and 5 HEPES, pH 7.4. Scale bar, 5 μm. (**k**) Representative $Ca^{2+}$ signals imaged from single dendrites (grey) and averaged traces (black) using GCaMP6f. (**l**) Maximum and averaged $\Delta F/F_0$ values of $Ca^{2+}$ signals in scrambled#1 and shRNA#1 condition. (**m**) Time course of averaged $\Delta F/F_0$ values and single exponential fits (red) of $Ca^{2+}$ signals in scrambled#1 and shRNA#1 condition. (**n**) Mean rise times and (**o**) decay τ of $Ca^{2+}$ signals in scrambled#1 and shRNA#1 condition n = 10–15; histograms are means ± s.e.m. *$P < 0.05$, **$P < 0.01$, ***$P < 0.001$; t-test.

using immunofluorescence (Fig. 2b, full neurons shown in Supplementary Fig. 3). The numbers and intensities of fluorescent puncta for all four markers were reduced in Shrm4-silenced neurons and rescued by HA-Shrm4 (Fig. 2c). The decrease occurred in parallel with a reduction in spine density and this was also rescued by HA-Shrm4 (Fig. 2d). These results were replicated by silencing via shRNA#2 (Supplementary Fig. 4a). Finally, to attribute these changes to Shrm4 specific knockdown, we silenced Shrm3 and found no changes to dendritic spine density as shown in Supplementary Fig. 2d.

Shrm4 silencing also had profound consequences on spine morphology reducing spine length, without affecting spine head width (Fig. 2e,f) while increasing the number of stubby spines, compared with scrambled controls (Fig. 2g). These effects were reversed using the rescue construct discounting non-specific effects of the shRNA (Fig. 2e–g). Interestingly, no changes in spine density were observed with Shrm4 knockdown after synaptogenesis at 12*DIV* (Supplementary Fig. 4b) suggesting a role for Shrm4 in synaptogenesis or in synapse maintenance.

To assess whether Shrm4 silencing reduced the density of spines via down-regulation of dendritic GABA$_B$Rs, we transfected hippocampal neurons with GABA$_{B1}$ shRNA, or scrambled shRNA[22] or GABA$_{B1}$ shRNA together with shRNA-insensitive GABA$_{B1b}$ cDNA before synaptogenesis (8$DIV$), for processing at 18$DIV$ for immunofluorescence. GABA$_{B1}$ knockdown reduced GABA$_{B1}$ immunofluorescence (Supplementary Fig. 5a), and similar to Shrm4 silencing, also reduced the density of spines, which was rescued by shRNA-insensitive GABA$_{B1b}$ cDNA (Fig. 2h).

Then, we evaluated the specific involvement of Shrm4–GABA$_B$Rs interaction on spine density by applying the Tat-859–870 peptide to hippocampal cultured neurons from 8$DIV$ to 13$DIV$ during the synaptogenesis peak. This treatment was sufficient to induce a reduction in spine density similar to that observed in Shrm4 or GABA$_{B1}$ knockdown condition demonstrating that the interaction between the two proteins is necessary for normal spine development (Fig. 2i). Interestingly, silencing of Shrm4 also resulted in changes in spontaneous dendritic Ca$^{2+}$-signals (Fig. 2j,k), assessed using GCaMP6f, without a change in mean amplitudes of spikes (Fig. 2l) in hippocampal neurons. In particular, we observed slower rise (Fig. 2m,n) and decay times (Fig. 2m,o) of Ca$^{2+}$ transients in Shrm4-knockdown neurons, which could have important consequences for spine morphology. Moreover, these defects were not rescued by the concomitant overexpression of GABA$_{B1}$ΔC mutant demonstrating the dependency of these effects on GABA$_B$Rs surface expression (Supplementary Fig 5b).

Thus, Shrm4 is localized at excitatory synapses and plays a crucial role in determining the morphology and molecular structure of dendritic spines most likely by modulating the surface expression of GABA$_{B1}$ receptors.

**Shrm4 and GABA$_{B1}$ are associated in a complex with DIC.** As Shrm4 knockdown reduced surface GABA$_{B1}$ receptors without affecting the total level of protein expression, we wondered whether this occurred because of a trafficking impairment. Immunostaining for GABA$_{B1}$ in permeabilized hippocampal neurons after Shrm4 knockdown revealed an increased intensity of intracellular GABA$_B$R expression in the soma compared with scrambled controls, indicating an accumulation in intracellular compartments which could be reversed by the rescue construct (Fig. 3a).

Accumulation of GABA$_B$Rs in the soma in Shrm4-silenced neurons could reflect increased internalization, or defective transport from the soma to the cell surface, or both. To distinguish, GABA$_B$R internalization was analysed in live hippocampal neurons[23]. These were transfected at 7$DIV$ with cDNAs encoding for GABA$_{B1a}$ tagged with a bungarotoxin (BTX) binding site, and GABA$_{B2}$, together with either knockdown or scrambled shRNAs. No difference in GABA$_B$R internalization rates between Shrm4-silenced neurons and control neurons were found (Supplementary Fig. 6a–e).

As a consequence, we investigated the transport of GABA$_B$Rs to distal compartments, which should be driven by microtubule-based motor proteins: kinesins and dyneins[24]. Kinesin-1 is responsible for GABA$_{B1a}$ transport through the axonal endoplasmic reticulum (ER) and ER–Golgi intermediate compartment (ERGIC[25]); while dynein targets GABA$_{B1}$ to the dendrites[26].

Using anti-Shrm4 antibody in adult rat brain lysates, we co-immunoprecipitated the dynein intermediate chain (DIC), GABA$_{B1}$ and GABA$_{B2}$, but not any of KIF5 isoforms (Fig. 3b; Supplementary Fig. 6f). We then confirmed the existence of this complex in transfected HEK293 cells by immunoprecipitating

Shrm4 with Shrm4-V5, and either GABA$_{B1a}$-myc or GABA$_{B1b}$-myc (Fig. 3c). Interestingly, DIC co-immunoprecipitated with HA-Shrm4 even in the absence of GABA$_{B1}$ suggesting that these two proteins are associated in a molecular complex (Fig. 3d).

We found histidine-tagged DIC associated with Shrm4-PDZ domain also in pull-down experiment *in vitro* demonstrating that the two proteins can interact (Fig. 3e). Of course, these results cannot exclude other factors, such as dynein light chains, that can participate in a Shrm4-DIC complex *in vivo*. Nevertheless, to identify which domain of Shrm4 was responsible for the interaction, we used a GST-tagged PDZ domain mutated in the classical PDZ binding site (K26G27/AA). This mutant was still able to pull-down histidine-tagged DIC *in vitro* (Fig. 3e) suggesting that another region of the Shrm4-PDZ domain was involved. The analysis of the Shrm4-PDZ solution structure (Protein Data Bank (PDB) code: 2EDP) highlighted the N-terminal segment (first 14 amino acids) as one of the portions more likely involved in interaction with DIC. Indeed, this segment of Shrm4-PDZ is solvent accessible and is opposite to the PDZ region that participates in classic protein's C-termini recognition (β2–α2 interface[27]). We truncated 14 amino acids at the N terminus of Shrm4-PDZ (GST-tagged PDZΔ14) and found that eliminating these residues completely abolished DIC binding in adult rat brain lysates (Fig. 3f). These results suggest that Shrm4 has the ability to bind GABA$_{B1}$ and DIC simultaneously, forming a ternary complex. Using *in silico* modelling, we propose a structure for this complex (Supplementary Fig. 6g). Collectively, the results of *in vitro* experiments suggest that the β1 and β2 strands of Shrm4 bind the R$^{859}$ LITRGEWQSEA$^{870}$ segment of GABA$_{B1}$ and DIC, respectively. By combining the information from our *in vitro* experiments with the structure availability of the Shrm4-PDZ, DIC 110–138 fragment in complex with LCs, and the coil–coil C-terminal heterodimer of GABA$_{B1b}$ and GABA$_{B2}$ (PDB: 4PAS[28]), we could predict a low resolution model of the Shrm4-PDZ in complex with the GABA$_{B1b}$–GABA$_{B2}$ heterodimer, DIC and LC8 (Supplementary Fig. 6g).

By using dSTORM on 18$DIV$ hippocampal neurons, we detected Shrm4 puncta on tubulin-positive filaments (Fig. 3g) along dendrites where Shrm4 could associate with DIC and GABA$_{B1}$.

Finally, to ascertain if Shrm4, DIC and GABA$_B$Rs associate in a physiological context, we generated adeno-associated virus serotype 5 (AAV5) expressing either scrambled (AAV5-scrambled#1) or Shrm4 knockdown (AAV5-shRNA#1) shRNAs and injected these into rat CA1 hippocampi (Fig. 3h). Using western blots, the AAV5-knockdown-injected hemisphere exhibited markedly lower Shrm4 protein levels than the AAV5-scrambled-injected hemisphere (Supplementary Fig. 6h). Subsequent anti-DIC immunoprecipitation from hippocampal lysates, previously injected with either AAV5-scrambled#1 or AAV5-shRNA#1, revealed co-immunoprecipitation of GABA$_B$R with DIC in scrambled controls; however, this was significantly reduced in lysates from the Shrm4-knockdown hemisphere suggesting that dynein–GABA$_B$R co-association is dependent on endogenous levels of Shrm4 (Fig. 3i). Taken together, these results highlight the central role played by Shrm4 in the association of dynein and GABA$_B$Rs in the hippocampus.

**Shrm4 mediates GABA$_B$R dynein-dependent dendritic transport.** As dynein transports postsynaptic proteins and Golgi outposts[29], and since Shrm4 orchestrates GABA$_B$R–dynein interaction, we postulated that Shrm4 could mediate the dynein-dependent GABA$_B$R trafficking to dendrites. To investigate, we analysed the

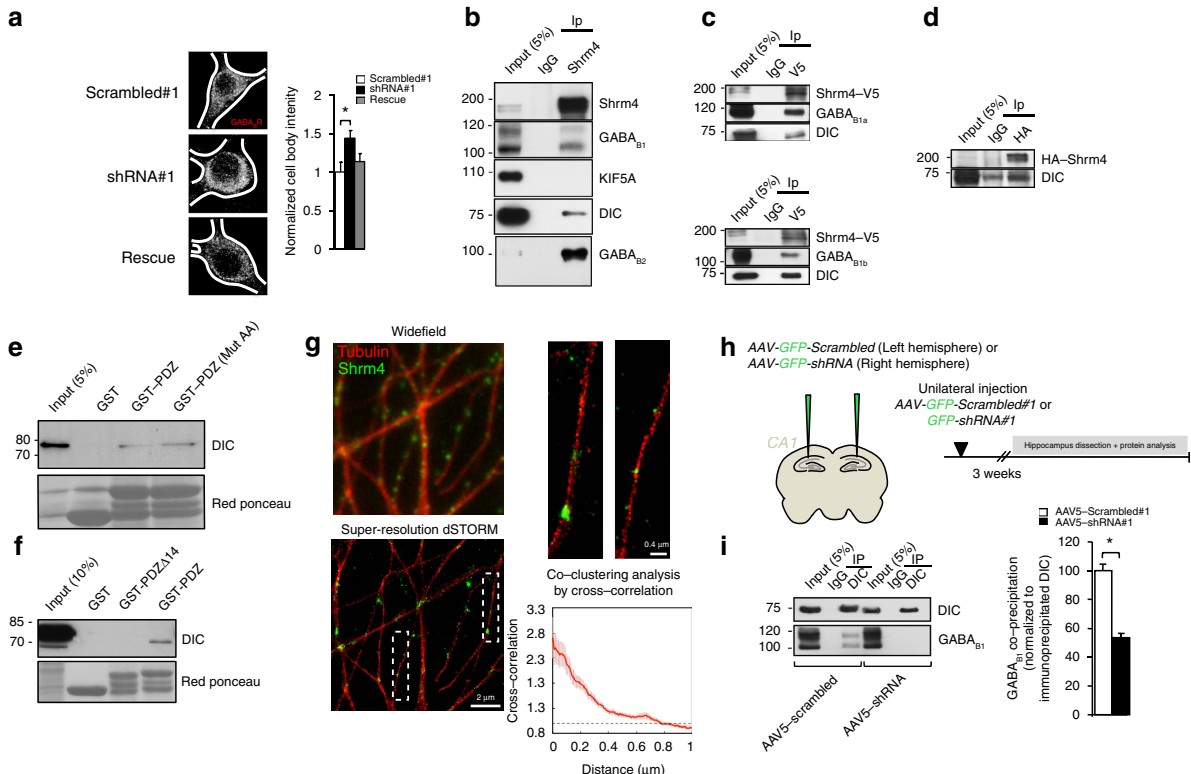

**Figure 3 | Shrm4 links GABA$_B$Rs to the dynein/dynactin complex.** (**a**) Immunofluorescence of GABA$_B$R intensity in the cell body of 18*DIV* cultured hippocampal neurons transfected with Shrm4 scrambled#1, knockdown (shRNA#1) or rescue constructs at 8*DIV*. All data in bar graphs throughout are means ± s.e.m.; $n = 10, 10, 15$ *$P = 0.0057$; F(2,32) = 6.094, one-way ANOVA-Mann–Whitney. (**b**) Co-immunoprecipitation experiments on brain extracts using polyclonal anti-Shrm4 show that Shrm4, GABA$_{B1}$, GABA$_{B2}$ and DIC are associated. By contrast, KIF5A is not present in this complex. (**c**) Co-immunoprecipitation of both GABA$_{B1}$ isoforms in HEK293 cells transfected with Shrm4-V5 and GABA$_{B1a}$-myc (top) or GABA$_{B1b}$-myc (bottom). GABA$_{B1a}$, GABA$_{B1b}$ and the endogenous DIC co-immunoprecipitated with Shrm4-V5. (**d**) Endogenous DIC co-immunoprecipitates with expressed HA-Shrm4 in HEK293 cells in the absence of GABA$_{B1a}$ or GABA$_{B1b}$ subunits. Monoclonal antibodies (anti-V5, anti-myc and anti-DIC) were used for western blot. (**e**) Western blot from *in vitro* pull-down assay of purified Histidine-tagged DIC using GST-PDZ and mutant GST-PDZ (AA). The Shrm4-PDZ domain is able to directly bind DIC even when its common binding site is mutated. Ponceau Red staining of the blot has been presented at the bottom. (**f**) GST pull-down experiments on brain extracts show that the Shrm4-PDZ domain pulls down DIC while the mutant GST-PDZΔ14 does not. Ponceau red staining of the blot has been presented at the bottom (full blot in Supplementary Fig. 11). (**g**) Conventional (top left) and super-resolution direct stochastic optical reconstruction microscopy (dSTORM) (bottom left) imaging of Tubulin-ATTO 488 (shown in red) and Shrm4-Alexa 647 (shown in green). Shrm4 positive puncta are localized along microtubule-positive filaments, as evidenced in the details of the super-resolution image (top right). The super-resolution data has been quantified by cross-correlation analysis. The positive, higher than 1 cross-correlation, indicates co-clustering of the two fluorescent signals (Error-bars are s.e.m.s, number of regions for the cross-correlation measurement: 30, number of fields: 3. Scale bar, 0.4 μm). (**h**) Schematic illustrating unilateral local injection of AAV5-scrambled#1 (left hemisphere) and AAV5-shRNA#1 (right hemisphere) into rat brain CA1 hippocampus, with time line for recovery and experiments. (**i**) Western blots and histograms showing monoclonal anti-DIC immunoprecipitation from scrambled and shRNA lysates of infected hippocampus extracts ($n = 6$ rats). Co-precipitated GABA$_{B1}$ levels were normalized to DIC immunoprecipitation levels and the normalized percentage of GABA$_{B1}$ co-precipitation was calculated. Histograms show mean ± s.e.m.; *$P < 0.05$, t-test.

distribution of endogenous GABA$_B$Rs in the dendrites and axons of 8*DIV* hippocampal neurons expressing: GFP (control), GFP-coexpressing knockdown shRNA#1 either with or without rescue-shRNA, and a GFP-tagged dominant-negative dynactin construct (GFP-p150-cc1) that inhibits dynein activity[30].

GABA$_B$R fluorescence intensity in dendrites of GFP-shRNA#1 and GFP-p150-cc1-expressing neurons was markedly lower compared with GFP controls (Fig. 4a), while fluorescence in axons was unaffected ($P > 0.05$). Expressing the rescue construct in Shrm4-silenced neurons recovered the GABA$_B$R fluorescence in dendrites (Fig. 4a). We also quantified the polarity index (PI), which has a higher value the greater the abundance of receptors in dendrites (Fig. 4a, full neurons are shown in Supplementary Fig. 7a). Interestingly, by expressing shRNA#1 in neurons at 12*DIV*, the PI was significantly lower compared with controls, suggesting that Shrm4 is

important for the normal abundance of dendritic GABA$_B$Rs not only before, but also after synaptogenesis (Supplementary Fig. 7b). Finally, a second GFP-Shrm4 knockdown construct (shRNA#2) also reduced the GABA$_B$R PI as shown in Supplementary Fig. 4a.

Since both GABA$_{B1a}$ and GABA$_{B1b}$ are present in dendrites[31], we considered if Shrm4 knockdown or dynein inhibition differentially modified their intracellular transport. Then we expressed shRNA#1 or GFP-p150-cc1 that significantly reduced the PI of expressed GABA$_{B1a}$ and GABA$_{B1b}$ subunits, consistent with the reduced intensity noted for endogenous GABA$_B$Rs (Fig. 4b, full neurons are shown in Supplementary Fig. 8a). Importantly, these results demonstrate that Shrm4 acts through dynein to drive GABA$_B$Rs to dendrites in hippocampal neurons and overexpression of GABA$_{B1a}$ or GABA$_{B1b}$ does not rescue the reduction in PI.

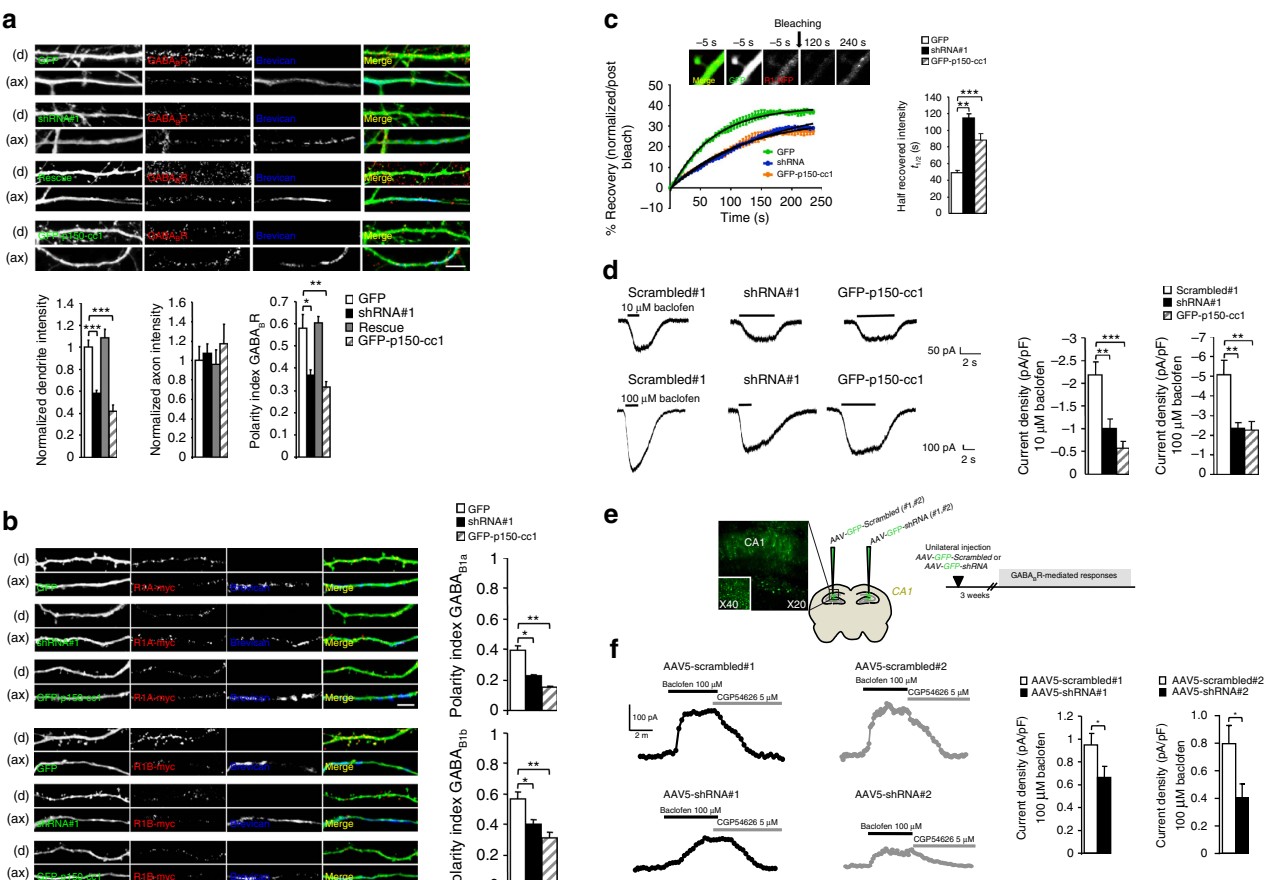

**Figure 4 | Association of Shrm4 with dynein/dynactin drives GABA$_B$Rs dendritic transport. (a)** Images of dendrites (d) and axons (ax) of hippocampal neurons at 14*DIV*, transfected with GFP (control), knockdown (shRNA#1), rescue, or GFP-p150-cc1 at 8*DIV*. The mean fluorescence intensities of GABA$_B$Rs are shown with a polarity index (PI = $(I_d - I_a)/(I_d + I_a)$). For uniformly distribution PI = 0, whereas for dendritic polarization PI > 0, and for axonic polarization PI < 0. GABA$_B$R immunostaining was reduced in dendrites in knockdown and GFP-p150-cc1-expressing neurons compared with those for control and rescue neurons (Left histograms, $n = 10, 10, 15, 10$; ***$P < 0.0001$; $F(3,41) = 21.91$; one-way ANOVA-Mann–Whitney). In addition, the mean PI of shRNA#1- and GFP-p150-cc1-expressing neurons is also reduced (right histograms, $n = 10, 10, 15, 10$; ***$P < 0.0001$; $F(3,41) = 14.72$; One-way ANOVA-Mann–Whitney). Scale bar, 10 μm. **(b)** Images, mean fluorescences and PIs for dendrites (d) and axons (ax) of 14*DIV* hippocampal neurons co-transfected with GFP (control), knockdown- (shRNA#1) or GFP-p150-cc1 together with either GABA$_{B1a}$-myc or GABA$_{B1b}$-myc at 8*DIV*. GABA$_{B1}$ was detected with anti-myc and brevican identified the axons. Histograms show mean ± s.e.m.; GabaB1a: $n = 10$-15; ***$P < 0.0001$; $F(2,27) = 49.75$; one-way ANOVA-Mann–Whitney; GABAB1b: $n = 10$-15; ***$P = 0.0001$; $F(2, 27) = 12.53$; one-way ANOVA-Mann–Whitney. Scale bar, 10 μm. **(c)** Fluorescence recovery after photobleaching (FRAP) of cultured hippocampal neurons expressing GABA$_{B1}$-RFP together with GFP, shRNA#1 or GFP-p150-cc1. The mean half-time ($t_{1/2}$) for fluorescence recovery (fluorescent recovery post-bleach corrected to 0%) is significantly longer in knockdown and GFP-p150-cc1-expressing neurons than GFP control. Histograms show mean ± s.e.m.; $n = 7$; ***$P = 0.0002$; $F(2,15) = 16.43$; one-way ANOVA-Mann–Whitney. **(d)** Whole-cell GIRK currents recorded in response to GABA$_B$R activation by 10 (top) or 100 (bottom) μM baclofen, from rat hippocampal neurons at 14*DIV*, transfected either with scrambled, knockdown (shRNA#1) or GFP-p150-cc1 at 8*DIV*. Baclofen-activated (10 and 100 μM) GIRK current densities in Shrm4-silenced, dynein-inhibited neurons and scrambled shRNA neurons are shown. Histograms show mean ± s.e.m.; 10 μM: $n = 10, 10, 11$; ***$P = 0.0003$; $F(2,28) = 10.96$, one-way ANOVA-Mann–Whitney; 100 μM: $n = 10, 9, 12$; ***$P = 0007$; $F(2,28) = 9.466$). **(e)** Schematic of injection of AAV5-scrambled#1 (or #2) and AAV5-shRNA#1 (or #2) into opposing CA1 areas of rat, with time line for recovery and experiments. **(f)** Representative traces and histograms showing changes in GIRK and current densities (pA/pF) from CA1 pyramidal neurons evoked by baclofen (100 μM, horizontal bars) and blocked by the selective GABA$_B$R antagonist, CGP54626 (5 μM). Histograms show mean ± s.e.m.; $n = 10$; (left histogram shRNA#1: *$P = 0.0298$; *t*-test; right histogram shRNA#2: *$P = 0.0471$, *t*-test).

Furthermore, we verified that Shrm4 knockdown was not altering GABA$_{B1}$–GABA$_{B2}$ dimerization[32]. We therefore co-transfected GABA$_{B1}$-myc and GABA$_{B2}$-flag with either shRNA#1 or its scrambled control. This was followed by immunostaining for the receptors. The analysis revealed high correlation in both scrambled and Shrm4-knockdown conditions excluding dysfunctional receptor dimerization (Supplementary Fig. 8b).

To address whether the reduction in dendritic GABA$_B$Rs was a consequence of decreased transport, fluorescence recovery after photobleaching (FRAP) was applied to cultured hippocampal neurons coexpressing GABA$_{B1}$-RFP with GFP-shRNA#1 or GFP-

p150-cc1. For both, fluorescence recovery of GABA$_{B1}$-RFP was significantly delayed compared with neurons expressing the GFP control. Only an increased half-time for fluorescence recovery, without affecting the mobile fraction (plateau reached after 350 s), was observed in Shrm4-silenced and GFP-p150-cc1-expressing neurons, suggesting that transport of dendritic receptors, occurring either intracellularly or at the surface level, was severely impaired (Fig. 4c).

These results suggest that Shrm4 mediates dendritic transport of the GABA$_B$Rs heterodimer via its interaction with GABA$_{B1}$ and DIC.

**Physiological role of Shrm4 *in vitro* and *in vivo*.** GABA$_B$Rs activate G protein-coupled inwardly-rectifying K$^+$ channels, generating slow inhibitory postsynaptic currents (IPSCs)[33]. As Shrm4 regulates GABA$_B$R dendritic cell surface number, we examined K$^+$ currents induced by the GABA$_B$R agonist baclofen (10–100 μM) on 14*DIV* hippocampal neurons transfected at 7*DIV* with either GFP-coexpressing knockdown shRNA#1 or scrambled#1 or p150-cc1. Peak K$^+$ current densities (pA/pF) were lower in Shrm4-silenced and dynein-inhibited neurons compared with scrambled controls (Fig. 4d), consistent with a reduced number of dendritic GABA$_B$Rs. This reduced current was reversed by expressing shRNA-insensitive Shrm4 (Supplementary Fig. 9a). By contrast, K$^+$ currents activated by metabotropic glutamate receptors were unaffected by Shrm4 silencing and dynein inhibition (Supplementary Fig. 9b). GABA$_A$R-mediated miniature IPSCs (mIPSCs) were also unaffected by Shrm4 silencing (Supplementary Fig. 9c), indicating that Shrm4 and dynein/dynactin selectively regulate GABA$_B$R-mediated responses.

To assess if Shrm4 silencing affected the neurophysiology of GABA$_B$Rs, we injected AAV5-scrambled (left hemisphere) and AAV5-shRNA#1 (right hemisphere) into the hippocampal CA1 region of 3 months-old rats[33] (Fig. 4e). Input/output currents were measured to evaluate the basal excitatory transmission and we did not observe any statistically significant difference between the two conditions (Supplementary Fig. 10a). We also recorded field excitatory postsynaptic potentials (fEPSPs) in the apical dendritic layer of CA1 3 weeks later by inducing long-term potentiation (LTP) or long-term depression (LTD) via Schaffer collateral stimulation. LTP and LTD induction and maintenance were unaffected (Supplementary Fig. 10b,c) consistent with the unchanged PSD-95 intensity in brain slices from injected animals (Supplementary Fig. 10d). The apparent contradiction with the defect observed in dendritic spine number can be explained by noting that these animals were injected at three months of age, when the peak of synaptogenesis has passed. We know that Shrm4 knockdown in mature neurons did not affect spine number (Supplementary Fig. 5b), whereas it is still able to impair GABA$_{B1}$ trafficking (Supplementary Fig. 4b) as shown by the decreased polarity index.

Whole-cell K$^+$ currents evoked by baclofen in the CA1 of acute slices from injected animals were significantly reduced by Shrm4 knockdown compared with scrambled controls (Fig. 4f). The injection of AAV5-shRNA#2 induced similar reductions in K$^+$ current confirming the specificity of our results. Thus, Shrm4 silencing reduces functional GABA$_B$R responses *in vitro* and *in vivo*, in accord with a reduced number of surface dendritic GABA$_B$Rs.

**Shrm4 silencing *in vivo* affects hippocampal tonic inhibition.** Recent evidence suggests that GABA$_B$R activation enhances the conductance of extrasynaptic δ subunit-containing GABA$_A$Rs of dentate gyrus granule cells (DGGCs)[34–36]. As Shrm4 silencing reduced GABA$_B$R activity, we explored if this affected tonic inhibition in DGGCs.

Animals were injected with AAV5-scrambled (left hemisphere) and AAV5-knockdown (right hemisphere) shRNAs into the DG (Fig. 5a). Whole-cell recordings 3 weeks later from DGGCs in acute slices[35,36] subject to scrambled shRNA revealed that GABA (5 μM) increased the bicuculline-sensitive baseline current and current noise variance (Fig. 5b). For cells expressing Shrm4-shRNA#1, both current and noise variance were reduced compared with controls (Fig. 5b). As tonic inhibition in DGGCs relies on δ subunit-containing GABA$_A$Rs[34] and can be regulated by GABA$_B$Rs, we considered whether the reduced tonic current

was a direct consequence of a reduction in surface GABA$_B$Rs. Repeating these experiments in the presence of the GABA$_B$R antagonist CGP54628, showed similar reduction in tonic current and noise variance evident in the knockdown condition without CGP pre-treatment (Fig. 5c). These data suggested that reducing cell surface GABA$_B$Rs has no direct effect on tonic inhibition, which is more likely due to an indirect impairment of δ subunit-containing GABA$_A$Rs. This was confirmed using the agonist THIP at δ subunit-selective concentrations (3 μM) and as expected, the mean current and noise variance were reduced in DGGCs in the hemisphere carrying the Shrm4 knockdown compared with the control hemisphere (Fig. 5d). To exclude off-target effects of shRNA#1, we injected a second knockdown construct, AAV5-shRNA#2, which produced identical effects on THIP-induced currents (Fig. 5e). By contrast, GABA$_A$R-mediated mIPSCs in DGGCs were unaffected (Supplementary Fig. 10e), indicating that Shrm4 specifically regulates δ subunit-containing GABA$_A$R-mediated responses.

Interestingly, co-immunoprecipitation using brain extracts and monoclonal anti-GABA$_A$R δ subunit revealed that GABA$_B$Rs and δ-subunit-containing GABA$_A$Rs co-associate. By contrast, the synaptic GABA$_A$R γ2 subunit[37] was absent (Fig. 5f). Thus, by controlling postsynaptic GABA$_B$Rs, Shrm4 is also able to regulate tonic inhibition mediated by δ subunit-containing GABA$_A$Rs.

**In vivo Shrm4 silencing causes behavioural deficits.** A role for GABA$_B$Rs in anxiety[38] and epilepsy[14,39] is well-known. GABA transmission has been also linked to neurodevelopmental disorders such as autism spectrum disorders (ASD)[40,41]. Furthermore, GABA$_B$R agonist application has been proposed as therapeutic strategy for social deficits, repetitive behaviours and other aspects of ASD in different mouse models[42]. To understand if reducing GABA$_B$R numbers and tonic inhibition by Shrm4 knockdown has behavioural implications, we injected rats bilaterally with either Shrm4 knockdown (AAV5-shRNA#1 or AAV5-shRNA#2), or AAV5-scrambled shRNA (AAV5-scrambled#1 or AAV5-scrambled#2) and extensively analysed the behaviour of these animals (Fig. 6a). Locomotor activity of the injected animals was unaffected allowing us to perform subsequent behavioural analyses without locomotor bias (Fig. 6b).

The elevated plus maze (EPM) and the marble-burying test measured anxiety levels[43] whereas social behaviours were evaluated in a three-chamber apparatus[44] and in the tube test for aggressivity[45]. AAV5-knockdown-shRNA-injected rats (with either shRNA#1 or shRNA#2) exhibited increased anxiety and impaired social behaviour (Fig. 6c–f). In the EPM, Shrm4-knockdown rats made fewer open-arm entries (Fig. 6c) and spent less time in open arms compared with AAV5-scrambled shRNA-injected controls (Fig. 6c). The total number of arm entries was unaffected by AAV5-knockdown-shRNA, confirming that locomotion was unaffected.

In the marble-burying test, animals injected with AAV5-shRNA#1 buried a higher number of marbles and spent less time before burying compared with AAV5-scrambled controls confirming an increased anxiety level (Fig. 6d). Social behaviours of animals injected with AAV5-shRNA#1 were also found impaired with reduced time spent close to a stranger naive animal (Fig. 6e, sociability) and to a second new stranger animal (Fig. 6e, social novelty) compared with AAV5-scrambled controls. These animals also showed greater aggression in the tube test in terms of percentage of wins versus the control-injected animals (Fig. 6f).

We then assessed involvement of Shrm4's deficiency in epilepsy by pentylenetetrazole (PTZ) administration to evaluate

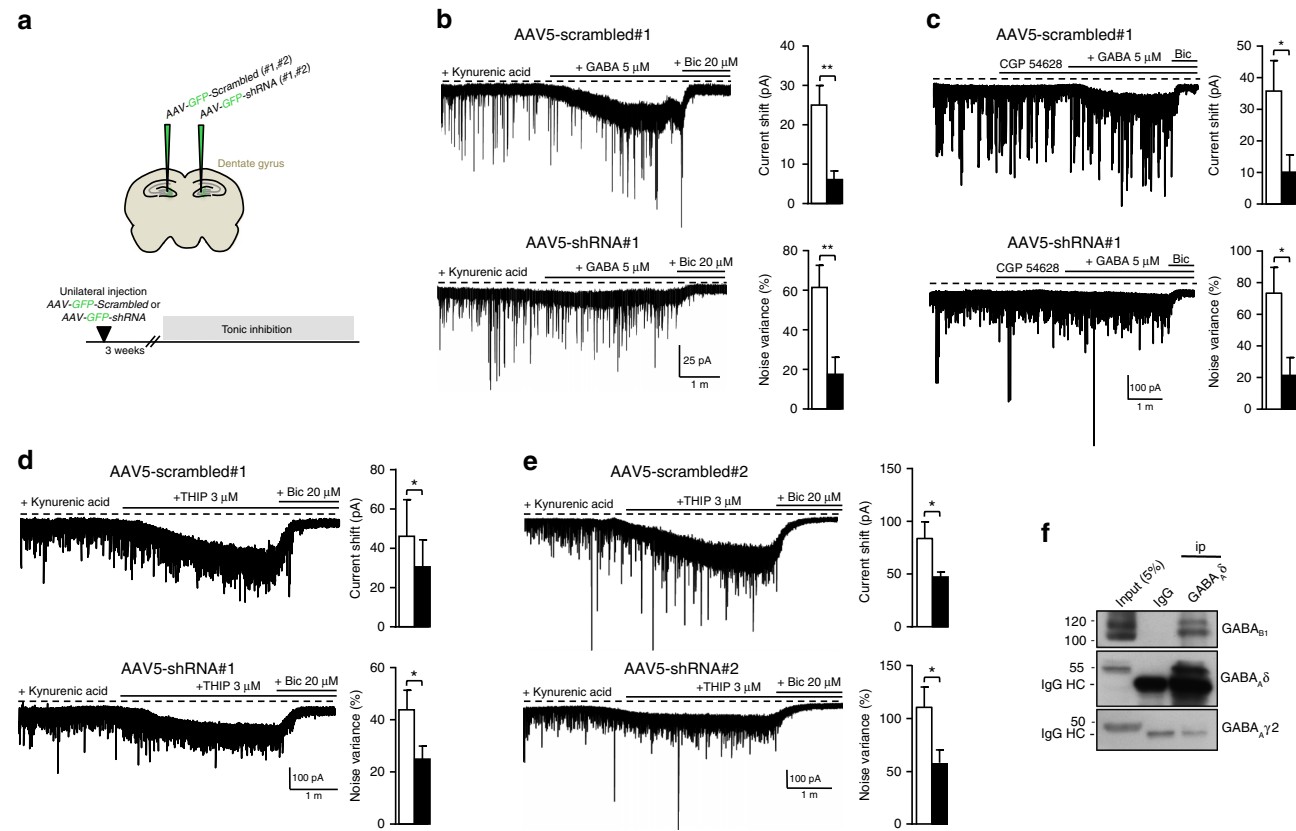

**Figure 5 | Shrm4 silencing in vivo mediates tonic inhibition in the hippocampus.** (**a**) Schematic of injection of AAV5-scrambled#1 (or #2) and AAV5-shRNA#1 (or #2) into opposing DG of rat, with time line for recovery and experiments. Slices were prepared 3 weeks after injection. (**b**) Tonic currents activated by 5 μM GABA in the presence of kynurenic acid (3 mM) of DG granule cells (DGGCs) in acute brain slices taken from hemispheres injected with either AAV5-shRNA#1 or AAV5-scrambled#1. GABA-mediated current shift and noise variance were lower in slices injected with AAV5-shRNA#1 compared with AAV5-scrambled#1 (current shift: $n = 10, 12$; $**P = 0.0050$; $t$-test; noise variance: $n = 9, 11$; $**P = 0.025$; $t$-test). (**c**) Tonic currents activated by 5 μM GABA in the presence of CGP54626 (5 μM) of DGGCs in acute brain slices taken from hemispheres injected with either AAV5-shRNA#1 or AAV5-scrambled#1. A lower GABA-mediated current shift and current variance in slices pre-treated with CGP54268 from AAV5-shRNA#1 compared with AAV5-scrambled#1 injected animals were observed (current shift: $n = 6, 7$; $*P = 0.0352$; $t$-test; noise variance: $n = 5, 7$; $*P = 0.0215$; $t$-test). (**d**) THIP (3 μM)-activated currents of DGGCs in the presence of kynurenic acid (3 mM). THIP current shift and current variance were lower in slices injected with AAV5-shRNA#1 compared with AAV5-scrambled#1 (current shift: $n = 9, 11$; $*P = 0.0225$; $t$-test; noise variance: $n = 10, 13$; $*P = 0.0419$; $t$-test). (**e**) THIP (3 μM)-activated currents from DGGCs in the presence of kynurenic acid (3 mM). A lower THIP current shift and current variance was also observed in slices injected with a second knockdown construct AAV5-shRNA#2 compared with AAV5-scrambled#2. Application of bicuculline (20 μM) demonstrates the GABA$_A$R-mediated specificity of the THIP current (current shift: $n = 9, 8$; $*P = 0.0341$; $t$-test; noise variance: $n = 10, 8$; $*P = 0.0465$; $t$-test). (**f**) Co-immunoprecipitation of GABA$_B$Rs and δ but not γ2 subunits of GABA$_A$Rs using brain extracts (full blot is shown in Supplementary Fig. 11) All histograms were presented as mean ± s.e.m.

seizure sensitivity[46] and recorded electro-encephalograms (EEG) to measure spontaneous electrical activity (Fig. 7a).

Electrical activity evaluated for 24 h in freely moving awake animals showed a significant spontaneous spike activity in all the AAV5-shRNA#1 injected rats compared with the AAV5-scrambled#1 (Fig. 7b,c) suggesting an increased general excitability.

In fact, following PTZ injections (i/p 45 mg kg$^{-1}$)[46], the latency to the first seizure was reduced and the seizure duration was longer for knockdown shRNA-injected animals (Fig. 7d). The severity of the response to PTZ was also increased with greater number of tonic-clonic seizures (Fig. 7d). This indicates that in vivo Shrm4 silencing in CA1 produces defects in anxiety, social behaviour and susceptibility to seizures. Similar effects on seizure susceptibility were obtained when we injected rats intraperitoneally with Tat-859–870 peptide administering PTZ 12 h after (Fig. 7e,f) demonstrating that the disruption of Shrm4–GABA$_B$Rs interaction was responsible of these effects.

These phenotypes parallel the defects in GABA$_B$R trafficking and transmission observed in vitro and in vivo in Shrm4-silenced hippocampal neurons.

## Discussion

Several neurodevelopmental IDs including autism and Fragile X syndrome are characterized by a reduction of GABA$_B$R expression levels[47,48], and treatment with GABA$_B$R agonists have been reported to improve susceptibility to seizures[49] and social and cognitive behaviour[50–52]. In this study, we have characterized an interaction between the ID-linked protein Shrm4, GABA$_B$Rs and dynein motor protein. The disruption of this complex leads to dysfunction of GABA$_B$Rs cell surface targeting and subsequent reduction of signalling efficacy, and this has interesting parallels with these neurodevelopmental disorders.

We have characterized physiological roles for Shrm4 by discovering a new interaction between its PDZ domain and the

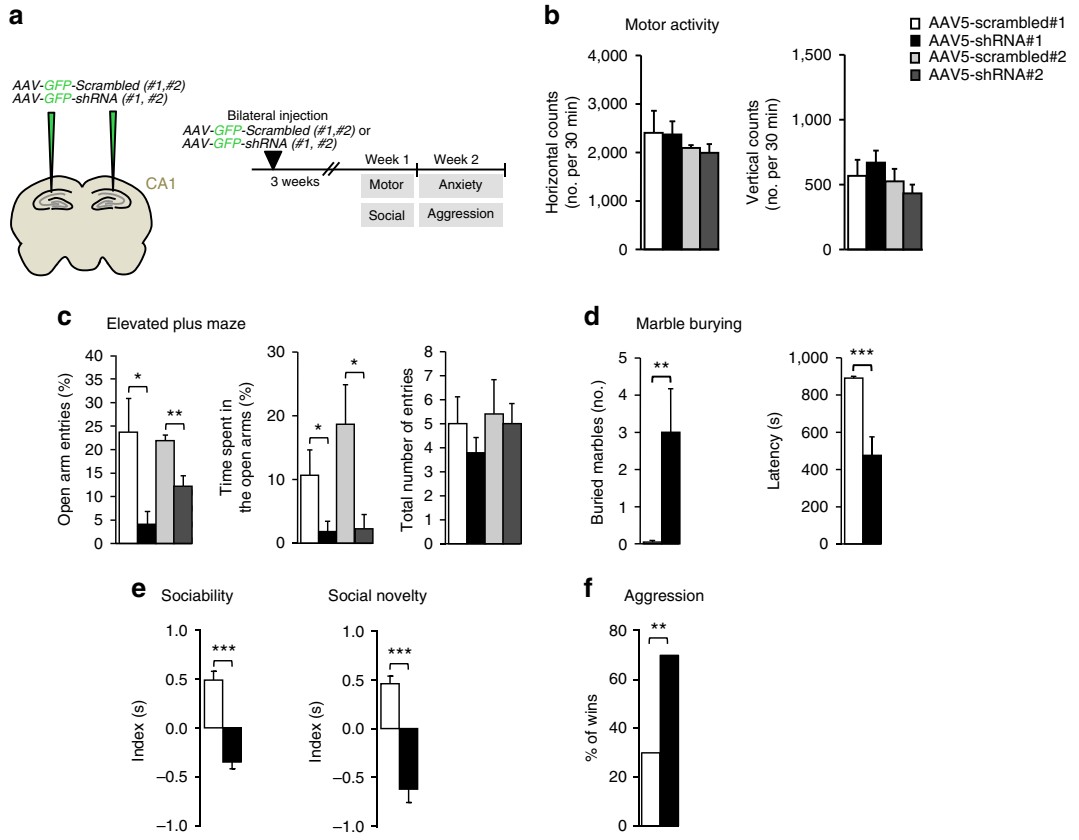

**Figure 6 | Shrm4 silencing *in vivo* increases anxiety and impaired social behaviours.** (**a**) Experimental design for investigating behavioural effects of Shrm4 silencing in rat hippocampal CA1. Rats were injected bilaterally with either AAV5-scrambled#1 (or #2) or AAV5-shRNA#1 (or #2) and tested after 3 weeks. (**b**) No change in motor function in terms of total number of counts evaluated in an automated activity cage for 30 min between Shrm4 knockdown or control animals. (**c**) A reduced number of open-arm entries and time spent in the open arms for Shrm4 knockdown; no difference in the total number of entries (open-arm entries: Scrambled#1 versus shRNA#1: $n = 12, 8$; $*P = 0.045$; $t$-test; Scrambled#2 versus shRNA#2: $n = 5, 5$; $**P = 0.005$; $t$-test) (time spent in the open arms: Scrambled#1 versus shRNA#1: $n = 8, 8$; $*P = 0.0307$; $t$-test; Scrambled#2 versus shRNA#2: $n = 8, 8$; $*P = 0.0133$; $t$-test). (**d**) An increased number of marbles buried and a decreased latency to the first burial were observed in knockdown animals (buried marbles: $n = 11, 10$; $**P = 0.008$; $t$-test) (latency: $n = 10, 10$; $***P = 0.0003$; $t$-test). (**e**) In the sociability and social novelty test, a reduced time close to stranger 1 in the sociability test and to stranger 2 in the social novelty test was observed in AAV5-shRNA#1 rats compared with AAV5-scrambled#1 rats; (Sociability: $n = 5, 5$; $***P < 0.0001$; $t$-test; social novelty: $n = 5, 5$; $***P = 0.0002$; $t$-test). All histograms were presented as mean ± s.e.m. (**f**) Tube test—AAV5-shRNA#1 rats showed a greater aggression compared with AAV5-scrambled#1 rats. $n = 5$; $**P = 0.0041$; Fisher's exact probability test.

GABA$_{B1}$ C terminus, a region important for GABA$_B$Rs trafficking[19,20]. Only one other PDZ domain-containing protein, Mupp1, is known to interact with GABA$_B$Rs, but its physiological role remains unknown[53]. The localization of Shrm4 to microtubule-positive filaments in dendrites suggests its interaction with GABA$_B$Rs is potentially critical for delivering these receptors to membrane-delimited signalling domains. Consistent with this, using shRNA silencing to reduce Shrm4 levels we observed decreased GABA$_B$R in dendrites, causing their accumulation in the soma. This most likely involved dysfunctional GABA$_B$R transport to the dendrites, a process driven by microtubule-based motor proteins: kinesins and dyneins. In this regard, we found that Shrm4 directly binds the dynein intermediate chain with a distinct portion of the PDZ domain (that is, the β1 and β2 strands) allowing for the simultaneous binding of Shrm4 with both GABA$_{B1}$ and DIC. In fact we found that GABA$_B$Rs and the dynein intermediate chain co-associate, and significantly, dynein inhibition reduced GABA$_B$R transport to the dendrites, but had no effect on trafficking into axons. Therefore, both dynein and Shrm4 are required to target GABA$_B$Rs to dendrites.

We know that Shrm4 can associate with both GABA$_{B1a}$ and GABA$_{B1b}$, and that Shrm4 silencing reduced the levels of both isoforms in the dendrites. However, GABA$_{B1a}$ is preferentially sorted to axons under physiological conditions[31] being targeted via its Sushi domains in pre-Golgi ER or ERGIC[25]. We surmised that only GABA$_{B1a}$ subunits that escape axonal targeting in pre-Golgi compartments can then associate with Shrm4 in the Golgi to be re-directed (like GABA$_{B1b}$) to dendrites[54].

On the basis of this premise, we propose that Shrm4, similar to the ID protein PQBP1, functions as an adaptor protein[55] for intracellular trafficking of cargo, by binding the C termini of GABA$_{B1}$ and also dynein to target these receptors to dendrites. This is further supported by our finding that *in vivo* Shrm4 silencing reduced the association between GABA$_B$Rs and dynein and differs from other post-translational modifications that influence the kinetics and pharmacological properties of GABA$_B$Rs[12].

Our biochemical and imaging data strongly suggest that Shrm4 and dynein play crucial roles in inhibitory transmission. Shrm4 silencing or dynein/dynactin inhibition specifically reduced baclofen-activated K$^+$ currents due to dysfunctional GABA$_B$R

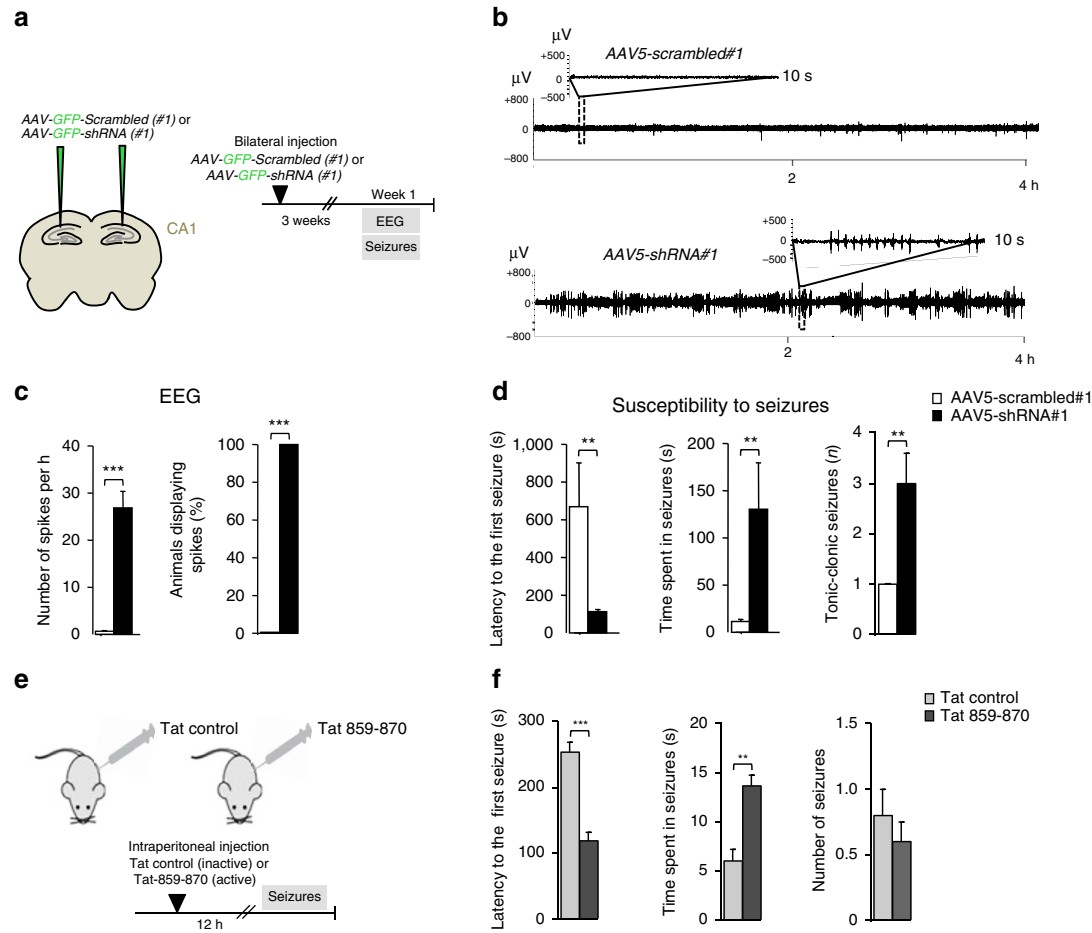

**Figure 7 | Shrm4 silencing *in vivo* increases neuronal excitability.** (**a**) Experimental design for investigating behavioural effects of Shrm4 silencing in rat hippocampal CA1. Rats were injected bilaterally with either AAV5-scrambled#1 or AAV5-shRNA#1 and tested after 3 weeks. (**b**) EEG recording in basal condition—two representative traces (4 h recording) of basal cerebral activity in freely moving awake rats are shown. The traces of the AAV5-shRNA#1 rat showed abnormalities in term of presence of spikes of high amplitude compared with AAV5-scrambled#1 rat. The traces of the remaining AAV5-shRNA#1 rats showed similar patterns (data not shown). (**c**) Quantification shows higher number of spikes and an increased number of animals showing spikes in AAV5-shRNA#1-injected rats compared with AAV5-scrambled#1 rats. (number of spikes/hour: $n = 5, 5$; ***$P < 0.0001$; $t$-test; animals displaying spikes: $n = 5, 5$; ***$P < 0.0001$; $t$-test). (**d**) PTZ injection ($45 \, \text{mg kg}^{-1}$ i.p.) significantly decreased the mean latency to the first seizure and increased the mean duration of seizures and number of tonic-clonic (Racine scale 5) seizures evaluated for 30 min in AAV5-shRNA#1 compared with AAV5-scrambled#1-injected rats (Latency to the first seizure: $n = 8, 9$; **$P = 0.0201$; $t$-test; time spent in seizure: $n = 9, 8$; ***$P = 0.0214$; $t$-test; number of tonic-clonic seizures: $n = 9, 8$; **$P = 0.0069$; $t$-test). (**e**) Experimental design for investigating the effect of intraperitoneal injection of the Tat-859–870 (active) peptide compared with the Tat control (inactive) on susceptibility to seizures. (**f**) The injection of the tat-859–870 given 12 h before the test significantly increased the susceptibility to PTZ-induced seizures ($45 \, \text{mg kg}^{-1}$ i.p.) in terms of reduced latency to the 1st seizure (***$P < 0.001$, $n = 3$; $t$-test) and increased time spent in seizures (***$P = 0.0057$, $n = 3$; $t$-test) even if the number of seizures was not different. All histograms show mean ± s.e.m.; ***$P < 0.001$; $n = 3$; $t$-test.

trafficking and depletion of surface dendritic $GABA_BRs$ without affecting $GABA_AR$-mediated mIPSCs and glutamate-mediated $K^+$ currents.

Depleting postsynaptic $GABA_BRs$ *in vivo* increased seizure susceptibility, as noted for $GABA_{B1}^{-/-}$ knockout mice[56]; the weaver mouse (with mutated GIRK2 channel), and the $Girk2^{-/-}$ null mouse, all of which are characterized by a significant loss of $GABA_BR$-mediated inhibition[53,57]. Thus it is not surprising that Shrm4 silencing, or the disruption of Shrm4–$GABA_BRs$ interaction, are associated with augmented neuronal excitability, as showed by the increase in spontaneous spikes recorded by EEG and in seizures susceptibility after PTZ injection.

$GABA_BR$ involvement in anxiety-related disorders remains unclear. While $GABA_{B1}^{-/-}$ knockouts show increased anxiety[58], mice lacking either $GABA_{B1a}$ or $GABA_{B1b}$ isoforms are not anxious[40], possibly due to isoform compensation. However, hippocampal Shrm4 silencing was associated with anxiogenic

behaviour, and as this impairs trafficking of both $GABA_BR$ isoforms to dendrites, the increased anxiety is likely due to a reduction in postsynaptic $GABA_BR$ numbers. Social behaviours also appeared to be impaired on Shrm4 knockdown; this is not surprising since GABA transmission has been linked to ASD symptoms[40–42]. In addition, down-regulating $GABA_BRs$ by Shrm4 silencing surprisingly affected tonic inhibition. We found $GABA_BRs$ and δ subunit-containing $GABA_ARs$ co-associated, which is consistent with their extrasynaptic localization in the molecular layer of the DG[59,60]. By silencing Shrm4, tonic inhibition mediated by δ subunit-containing $GABA_ARs$ is impaired, which could also contribute towards increased seizures, a feature noted previously for $Gabrd^{-/-}$ mice[61].

Defects in dendritic spine density and shape are established pathological correlates of X-linked IDs[21,62] and Shrm4 silencing, before synaptogenesis, reduced spine density, length and affected their pre- and postsynaptic molecular composition. Shrm4 binds

F-actin and can influence actin remodelling in non-neuronal cells[4] and F-actin dynamic is crucial for synaptogenesis and spine plasticity[63,64]. Although Shrm4 could regulate the spine cytoskeleton, this is considered unlikely because Shrm4 silencing after maturation has no effect on spine morphology, even though dendritic $GABA_BR$ number were still reduced. This could explain why LTP induction and stability was unaffected by Shrm4 knockdown and is consistent with the maintenance of LTP in $GABA_{B1b}^{-/-}$ knockout mice, indicating that postsynaptic $GABA_BRs$ are not affecting LTP induction[31]. The dendritic spine defects might derive indirectly from impaired $GABA_BR$ trafficking caused by Shrm4 silencing. Indeed, $GABA_BR$ silencing before synaptogenesis, or the inhibition of Shrm4–GABAB interaction with Tat-859–870, induced a similar reduction in spine density. In addition, given $GABA_BRs$ modulate dendritic $Ca^{2+}$ signals[65], by inhibiting voltage-gated $Ca^{2+}$ channels, perturbation in $Ca^{2+}$ signalling due to altered trafficking of $GABA_BRs$ could also underlie spine defects.

In conclusion, our data identify Shrm4 as an important protein for synaptogenesis and for maintaining the inhibitory equilibrium mediated by the $GABA_BRs$ and extrasynaptic δ subunit-containing $GABA_ARs$. The consequences of disrupting Shrm4 expression are severe and manifest by increased anxiety, social behaviours impairments and a predisposition towards epilepsy.

## Methods

**Animals.** Experimental procedures were performed in accordance with the European Communities Council Directive (86/809/EEC) on the care and use of animals and the UK Animals (Scientific Procedures) Act 1986, and were approved by the CNR Institute of Neuroscience.

**cDNA, shRNA constructs and cell-permeable peptide.** The human KIAA1202 open reading frame 1 was subcloned into: pcDNA4/V5-HisB (Invitrogen) to obtain Shrm4-V5; pEGFP-C1 (Clontech) to obtain GFP-Shrm4; and pTL1-HA3 to obtain HA-Shrm4 (gift from Vera Kalscheuer). Flag-tagged $GABA_{B2}$, myc-tagged full-length $GABA_{B1a}$, $GABA_{B1a}$-Δ859–961, $GABA_{B1a}$-Δ922–961 and $GABA_{B1a}$-Δ870–961 were cloned into the pRK5 plasmid for expression in HEK293 cells to identify the minimal $GABA_{B1}$ sequence that interacts with Shrm4 (ref. 23). pCl-myc-$GABA_{B1a}$ and pCl-myc-$GABA_{B1b}$ constructs were used to investigate the distribution of $GABA_{B1a}$ and $GABA_{B1b}$ isoforms in cultured hippocampal neurons[20]. $GABA_{B1a}$-mRFP (generous gift by Andrés Couve, University of Chile) was used for fluorescent recovery after photobleaching (FRAP) experiments. Twenty-one base-pair small hairpin RNA sequences or their scrambled controls (shRNA#1—GCTCACGGTGTCGAAGATTGA (RAT-RESCUE) (nucleotides 22-45); shRNA#2—AGCAAAGGGAATCATTTAGTA (RAT + HUMAN) (nucleotides 3,221–3,241); shRNA#1 scrambled—GATTCTAGCCGGACGGTG TAA and shRNA#2 scrambled—GGTATAAGCAACGTAATGAAT) were subcloned into the vectors pLVTHM-GFP (shRNA#1 and shRNA#2 used in vitro experiments) and pAAV-U6-ZsGreenGFP (shRNA#1; in vivo experiments) (both from Penn Vector Core, Pennsylvania). The 21 base-pair sequences of the small hairpin RNAs (shRNA $GABA_{B1}$) for silencing $GABA_{B1}$ subunits was GAATCTGCTCCAAGTCTT and the scrambled sequence was GTGCTATTAC CCGTACAT[22].

Cell-permeable peptides (CPPs) were obtained from Primm (Italy) and China Peptide (China) (Tat control: YGRKKRRQRRR-TETWGQRSIE; Tat-859–870: YGRKKRRQRRR-ITRGEWQSET). Lyophilized CPPs were resuspended in sterile deionized water and used at the final concentration of 10 μM for pull-down treatments, 20 μM for hippocampal culture treatments whereas were administered by intraperitoneal injection at 3 nmol g$^{-1}$ in vivo[66].

The GW1-GFP-p150-cc1 construct was kindly donated by Casper Hoogenraad (Utrecht). GW1-GFP was used as control. For GST pull-down assays, the PDZ domain (aa 1–91), PDZΔ14 (aa 1–14) and mutated PDZ (KG/AA) of Shrm4 were subcloned into pGEX-4T-1 to create GST-PDZ, GST-PDZΔ14 and GST-PDZ (MutAA), respectively. For Y2H screens, Shrm4 DNA corresponding to (aa) 91–1,492 (ΔPDZ), 1–91 (PDZ) and 1,213–1,492 (ΔASD2) were subcloned into the pGBKT7 vector and used as bait. The His-tagged IC-2C (DIC) was kindly donated by Andy Catling (New Orleans). The Shrm3 siRNA sequence has been used previously[67].

**Cell cultures, transfection and lentiviral infection.** HEK293 (Thermo Fisher Scientific) cells at 50–70% confluence (24 h after plating) were transiently transfected with cDNA expression constructs (0.5–1 μg DNA per well in optiMEM, Invitrogen) using lipofectamine 2000 (Invitrogen) for 1 h in 5% CO$_2$ at 37 °C. Transfected cells were washed twice with PBS and grown for 48 h in Dulbecco's

modified Eagle medium supplemented with 10% fetal bovine serum and 1% penicillin/streptomycin, before fixation for immunocytochemistry, lysis for co-immunoprecipitation, or GST pull-down. The 293FT cell line was used to generate lentivirus and was grown in 1% G418 antibiotic. Primary hippocampal neurons were prepared from Wistar E18 rat brains[68,69] and plated onto coverslips coated with poly-D-lysine (0.25 mg ml$^{-1}$) at 75,000 per well for immunochemistry, or 300,000 per well for GST pull-down, co-immunoprecipitation and lentivirus infection. Neurons were transfected or infected at 7–8DIV or 12–13DIV and processed for experiments at 16–20DIV. For transfection, lipofectamine 2000 was used. Infection with Shrm4-shRNA or scrambled shRNA was performed as described previously[70].

**Western blots.** Proteins were transferred from gels onto nitrocellulose membranes followed by incubation with the following primary antibodies at room temperature for 2–3 h in 5% milk at the stated dilutions: Shrm4 (1:200, used previously[4]), GFP (1:2,000, MBL International), $GABA_A$α1 (1:1,000, Millipore), HA (1:1,000, Invitrogen) and $GABA_{B2}$ (1:1,000, gift of B.B.) KIF5A (1:1,000, Abcam), KIF5B (1:1,000, gift of F. Navone[71]) and KIF5C (1:1,000, Abcam) (all raised in rabbit); and: α-tubulin (1:40,000, Sigma), $GABA_B$ Receptor 1 (1:2,000, Abcam), $GABA_A$δ subunits (1:500, ROCKLAND), V5 (1:1,000, Invitrogen), myc (1:2,000, MBL International), Synaptophysin (1:500, NeuroMab), DIC (1:1,000, Abcam) (raised in mouse); and $GABA_A$γ2 (1:1,000, Abcam) raised in guinea pig. After rinsing, primary antibodies were revealed by incubation at room temperature for 1 h with horseradish peroxidase-conjugated anti-rabbit or anti-mouse antibodies (both 1:2,000 from GE Healthcare) and immunoreactive bands on blots were visualized by enhanced chemiluminescence (GE Healthcare).

**GST pull-down and immunoprecipitation.** GST fusion proteins were prepared in BL21 E. coli and purified using standard procedures. For co-immunoprecipitation, HEK293 cells, cultured neurons or rat brain homogenates (homogenization buffer: 50 mM TRIS–HCl, 200 mM NaCl, 1 mM EDTA, 1% NP40, 1% Triton X-100, pH 7.4, protease inhibitor cocktail) were centrifuged at 10,000g for 30 min at 4 °C, and supernatants incubated with antibodies (see below) at 4 °C overnight. Protein A-agarose beads (GE Healthcare, USA) were then incubated with the supernatants at 4 °C for 2 h. Beads were washed three times with lysis buffer, resuspended in 3 × sample buffer and after boiling for 5 min, the resulting solution was analysed by SDS–PAGE, followed by western blotting with antibodies (see below).

For in vitro interaction assays, His-tagged DIC was expressed in E. coli BL21 strain and purified with a Protino Ni-Ted Kit (Macherey-Nagel). Eluted DIC using imidazole gradient was incubated with GST-PDZ and mutant GST-PDZ (AA) for 3 h at room temperature, washed four times with PBS and analysed.

Antibodies and dilutions for co-immunoprecipitation were: Shrm4 (1:100, used as described in ref. 4), GFP (1:1,000, MBL International), and HA (1:1,000, Invitrogen) (raised in rabbit); $GABA_B$ receptor 1 (1:2,000, Abcam), $GABA_A$δ subunits (1:500, ROCKLAND), V5 (1:400, Invitrogen), and DIC (1:1,000, Abcam) (raised in mouse).

**Immunofluorescence and surface staining.** Cultured hippocampal neurons were fixed in 4% paraformaldehyde/4% sucrose for 10 min at room temperature and incubated in GDB1X solution (2 ×: gelatin 2%, Triton X-100 0.3%, 0.2 M Na$_2$HPO$_4$ pH 7.4, 4 M NaCl) for 2 h at room temperature with primary antibodies: Shrm4 (1:200, used previously[4]), synapsin 1 (1:200, SYnaptic SYstems) raised in rabbit; b-Tubulin (1:10,000, Sigma), PSD-95 (1:400, NeuroMab), GluA2 (1:200, NeuroMab), bassoon (1:100, Assay Designs), GABAB receptor 1 (directed against a surface epitope, 1:200, Abcam), V5 (1:500, Invitrogen), myc (1:1,000, MBL International), $GABA_A$ B3 (1:300, NeuroMab) (all raised in mice); Brevican (1:500, BD Transduction Laboratories) raised in guinea pig. A similar protocol has been used for dSTORM imaging.

For surface staining, live 18–20DIV hippocampal neurons were incubated for 10 min at 37 °C with antibodies against $GABA_BRs$. After washing (PBS supplemented with 1 mM MgCl$_2$ and 0.1 mM CaCl$_2$) neurons were fixed for 10 min at room temperature in 4% paraformaldehyde/4% sucrose without permeabilization. Coverslips were then washed, and incubated with Alexa 488 (1:400, Invitrogen), Alexa 555 (1:400, Invitrogen) or Cy5 (1:200, Jackson Laboratories) secondary antibodies for 1 h at room temperature.

**Colocalization analysis—Pearson correlation coefficient (r).** Pearson correlation coefficient (r) statistic was used to analyse the linear colocalization between fluorophores using the JACoP plugin of ImageJ. For two directly interacting proteins, the colocalization value of r tends towards 1. For high correlation, r is between 0.5 and 1; medium correlation: 0.3 and 0.5; low correlation: 0.1 and 0.3.

**Image acquisition.** Confocal images were obtained as described previously[72]. Briefly, fluorescence images were acquired with an LSM510 Meta confocal microscope (Carl Zeiss; gift from F. Monzino) and a × 63 objective (numerical aperture 1.4) with sequential acquisition setting, at 1,280 × 1,024 pixels resolution.

Image data were Z series projections of about 6–10 images, each averaged four times and taken at depth intervals of 0.75 µm.

**Quantification of GABA$_B$R fluorescence intensity.** Images were quantified as previously described in ref. 72. Images were acquired using a × 63 objective and average intensity of signals in proximal axon and primary dendrites were measured in ImageJ. Brevican, a proteoglycan present in the growth cone was used to identify the beginning of the axon and confirmed axonal identification. To prevent selection bias during quantification, the axon and dendritic segments were selected in one channel (GFP to visualization neuronal morphology) and quantified in the other channel (GABA$_B$Rs). A third channel was used to identified the axon (Brevican). The axon and dendritic signals were measured in segments of the same size. To control for background signals, we measured the intensity near the axon or dendrite (same segment size) and subtracted the random fluorescence intensity in these images. The average dendrite intensity $I_d$ and average axonal intensity $I_a$ was used to calculate the polarity index (PI) using PI $= (I_d - I_a)/(I_d + I_a)$. For uniformly distributed proteins $I_d = I_a$ and PI $= 0$, whereas PI $> 0$ or PI $<$ indicates polarization towards dendrites and axons, respectively[30].

**Calcium imaging and analysis.** Ca$^{2+}$ transients were imaged at 20 Hz in Krebs using GCaMP6 in hippocampal neurons transfected at 7DIV with scrambled, Shrm4 knockdown and GABAB1ΔC constructs along with cDNAs for GCaMP6f and dsRed. Fluorescence intensities ($F$) from ROIs, drawn around dendrites were normalized to baseline fluorescence ($F_0$) to obtain values of $\Delta F/F_0$. $\Delta F/F_0$ peaks were detected in Matlab using the Peakfinder plugin (Nathanael Yoder, Mathworks). Ca$^{2+}$ transients less than × 3 the signal-to-noise ratio were excluded from the analysis.

**dSTORM and analyse of cross-correlation.** Super-resolution localization imaging was carried out by direct stochastic optical reconstruction microscopy (dSTORM)[73]. Briefly, the fluorescent molecules of a sample prepared according to a standard immunofluorescence protocol (see above) were induced to blink on and off by modifying the chemical composition of the medium. If few molecules fluoresce at a given time, such that the diffraction limited spots corresponding to individual fluorophores are well separated, the position of each molecule can be quantified with higher accuracy than the resolution limit[74]. By acquiring multiple images of the same field (tens to hundreds of thousands), and storing the position of the localized molecules at each frame it is therefore possible to obtain an image of the sample with resolution higher than the diffraction limit. dSTORM was performed on an optical setup based on a Leica SR GSD-3D (Leica Microsystems Srl, Milan, Italy) super-resolution microscope equipped with a × 160 1.43 NA objective, an Andor iXon Ultra-897 EM-CCD sensor and three (405 nm 30 mW, 488 nm 300 mW and 642 nm 500 mW) solid state lasers. The sample was mounted on the stage and the medium was substituted before acquisition with a mix of glucose oxidase (560 µg ml$^{-1}$), caltalase (400 µg ml$^{-1}$) and Cysteamine HCl (100 mM) in TN buffer with 10% glucose w/v at pH 8 to induce blinking of the fluorophores[75]. Alexa 647 and Atto 488 were imaged sequentially, starting from the red channel. 30,000 images were collected for each channel with 5–20 ms exposure times and an increasing ramp of 405 nm laser intensity, with powers between 0 and 0.8 mW, to reactivate the molecules in long-lived dark states. The super-resolution images were reconstructed using the proprietary analysis software, discarding all the detected events with less than 20 photons/pixel for Alexa 647 and 40 photons for Atto 488. To quantify the co-clustering between the two labelled proteins, the list of particles detected for each channel were used to compute the spatial cross-correlation between the two signals[76], using previously published routines based on fast Fourier transformation written in Matlab[77]. Briefly flat cross-correlation curves are representative of no co-clustering between the two labelled proteins, while curves that significantly differ from 1 indicate that the two proteins cluster together. For each condition, the cross-correlation was evaluated on ∼30 randomly chosen square regions of 2 µm sides. For visualization purposes, super-resolution images were rendered as a 2D histograms with pixel sizes equal to 20 nm.

**Y2H screening.** For Y2H experiments, a fragment corresponding to the Shrm4 N-terminal PDZ domain (aa 1–91) was cloned in-frame with the GAL4 DNA-binding domain of the pGBKT7 vector and used as bait to screen an adult human brain cDNA library (Clontech, Mate and Plate Library). Positive yeast clones grew on plates containing X-GAL and Aureobasidin A (QDO/X/A plates) and expressed all four reporter genes: HIS3, ADE2, AUR1C and MEL1 under the control of three distinct Gal4-responsive promoters. cDNA plasmids from positive clones were recovered with the Easy Yeast plasmid isolation kit and transformed into DH5 E. coli grown on ampicillin plates, followed by sequencing.

**Cell surface biotinylation assay.** Hippocampal neuron membrane proteins were biotinylated using membrane-impermeable sulfo-NHS-SS-biotin (0.3 mg ml$^{-1}$, Pierce) for 5 min at 37 °C. The neurons were then washed with tris-buffered saline (TBS) supplemented with 0.1 mM CaCl$_2$, 1 mM MgCl$_2$ and 50 mM glycine at 37 °C, and rinsed with TBS supplemented with 0.1 mM CaCl$_2$ and 1 mM MgCl$_2$ (without

glycine) on ice, followed by lysis in extraction buffer (50 mM Tris–HCl, pH 7.4, 1 mM EDTA, 150 mM NaCl, 1% SDS, and protease inhibitors). Lysates were boiled for 5 min and biotinylated membrane proteins were precipitated with streptavidin-conjugated beads (Dynabeads, Invitrogen). The beads were washed with lysis buffer, boiled for 5 min in sample buffer, filtered, and the proteins in solution separated by SDS–PAGE for western blotting.

**Live-cell internalization assay.** GABA$_B$R internalization in live hippocampal neurons was investigated as described previously[78,79]. Neurons at 14–21DIV expressing GABA$_B$R heterodimers (R1a with bungarotoxin (BTX) binding site along with R2) and either GFP-shRNA or GFP-scrambled-shRNA were incubated in 1 mM d-tubocurarine for 2 min at room temperature followed by 3 µg ml$^{-1}$ BTX-AF555 (which attaches to the R1a binding site) for 10 min at room temperature. Receptor internalization was then followed as decay in surface fluorescence at 30–32 °C under the confocal microscope.

**Live-cell imaging and FRAP.** For live-cell imaging, neurons co-transfected with GABA$_{B1}$-RFP together with either GFP, Shrm4-shRNA or GFP-p150-cc1, were placed in an incubator with imaging medium at 37 °C and 5% CO$_2$ mounted on a Zeiss LSM510 Meta confocal microscope. Neurons expressing Shrm4-shRNA or scrambled shRNA were imaged with the 458 nm laser; neurons expressing GABA$_{B1}$-mRFP were imaged with the 543 nm laser.
FRAP experiments were performed following Fossati et al.[80] on GABA$_{B1}$-RFP-positive dendrites; regions of interest (ROI) on dendrites were defined and a pre-bleaching image acquired at the start. The ROI was next bleached by scanning 30 times with 405 and 458 nm lasers until no fluorescence signal was detectable. Fluorescence signal recovery was imaged over 10 min, and normalized to the total fluorescence of the pre-bleached ROI, which was verified as constant over time. The images were analysed with ImageJ and the results analysed with Prism (GraphPad).

**Computational modelling.** The structural models of GABA$_{B1b}$ and GABA$_{B2}$ were developed by comparative modelling using MODELLER[81]. We first modelled the 591–918 sequence GABA$_{B1b}$, comprising the seven-helix bundle, the intracellular and extracellular loops, and the C terminus up to the end of the coil–coil region (879–918 sequence) by using the crystal structure of the metabotropic glutamate receptor 5 (mGluR5; PDB 5CGC) as a template. One hundred models were generated by randomizing all the Cartesian coordinates of standard residues in the initial model. Extra-helical restraints were imposed to the following portions of the receptor: 766–772 and 879–918, corresponding, respectively, to the N terminus of helix 5 (H5) and the C terminal coil-coiled helix. In the model selected according to a corroborated procedure[82], the 887–918 portion of the C-terminal helix was replaced with the corresponding helix from the crystallographic coil–coil heterodimer (PDB: 4PAS). The best selected model of the GABA$_{B1b}$ served as a template to build the seven-helix bundle and the loops of the GABA$_{B2}$R. The best model among the 100 produced was docked onto the structural model of GABA$_{B1b}$R following an already described procedure[82]. The receptor orientation corresponding to an H1–H1, H7–H7 dimer, the most compatible with the C terminal coil–coil heterodimer, was employed for model completion by adding the C terminus. In this respect, the C-terminal helix was extracted from the crystallographic heterodimer (PDB: 4PAS), whereas the segment connecting the C-terminal end of H7 and the coil-coiled helix was built by means of the loop routine in MODELLER.
The information from our in vitro experiments were employed to drive the docking between the 859–870 GABA$_{B1b}$R and the β2 strand of Shrm4-PDZ and between the β1 strand of Shrm4-PDZ (PDB: 2EDP) and the 128–135 strand of IC2 in complex with dimeric LC8 of dynein (PDB: 2PG1). The two strands were docked so as to form an extension of the antiparallel sheet between IC2 and LC8.

**Injection of adeno-associated virus constructs.** Five to seven-weeks-old male Sprague Dawley rats, kept under 12 h light/dark conditions, with ad libitum food and water, were anaesthetized with zoletil$_{100}$ (0.1 ml per 100 g) and xylazine (0.1 ml per 100 g) and secured in a Kopf stereotaxic frame. A cannula was inserted into each CA1 (ref. 22). AAV5-shRNA#1 (or #2) or AAV5-scrambled shRNA#1 (or #2) (Penn Vector Core, University of Pennsylvania, USA) were injected directly into the CA1 (coordinates AP: − 3.8 mm relative to Bregma; L: + /− 2 mm; DV: − 2 mm from skull; Paxinos and Watson, 1985) or into the Dentate Gyrus (coordinates AP: − 3.6 mm relative to Bregma; L: 3.6 + /− mm; DV: − 3.5 mm from skull; Paxinos and Watson, 1985). Zoanthus sp. green fluorescence protein was also incorporated into the AAV construct to report injection coordinates and volumes. The animals were allowed to recover for 3 weeks before experimentation.

**Whole-cell patch-clamp recording of GIRK currents.** Current densities of G protein-coupled inwardly rectifying K$^+$ (GIRK) channel conductance in response to GABA$_B$ activation by baclofen, or mGluR activation by glutamate, were recorded from hippocampal neurons in culture at 12–16DIV, using whole-cell patch-clamp electrophysiology as described previously. Neurons were transfected at 7DIV with shRNA, scrambled shRNA or the coiled-coil domain of dynactin (GFP-p150$^{glued}$-

cc1), and the cells were identified by their expression of GFP. Rescue experiments were performed from neurons co-transfected at 7DIV with shRNA#1 and HA-Shrm4 and the cells were identified by their expression of GFP. Patch pipettes made of thin-walled borosilicate glass (outer diameter, 1.5 mm; inner diameter, 1.17 mm; GC-150TF-10; Harvard Apparatus, Kent, UK), (4–5 MΩ) were filled with internal solution (mM: 120 KCl, 2 MgCl$_2$, 11 EGTA, 30 KOH, 10 HEPES, 1 CaCl$_2$, 1 GTP, 2 ATP, pH 7.4). Neurons were continuously perfused with Krebs solution (mM): 140 NaCl, 2.5 CaCl$_2$, 4.7 KCl, 1.2 MgCl$_2$, 11 glucose, and 5 HEPES, pH 7.4 and voltage-clamped at −70 mV in whole-cell configuration, and series-resistance compensation applied. Currents for 20 sweeps of −10 mV hyperpolarising pulses were recorded with an Axopatch 200B amplifier (Molecular Devices, CA, USA) at 10 kHz filtering. Similar sweeps were recorded at regular intervals after applying series-resistance compensation of ~40%. Cells were discarded in the event of a change in series resistance >10%. To increase the size of GIRK currents and to convert them to inward currents, before baclofen or glutamate application, the KCl concentration was increased to 25 mM and the NaCl concentration reduced to 120 mM in the Krebs solution. This changed the $E_K$ from approximately −90 to −47 mV. In addition, 2 mM kynurenic acid and 20 μM bicuculline were added to the Krebs to block ionotropic glutamate and GABA receptors. Peak K$^+$ current amplitudes were filtered at 5 kHz before storage for analysis with Clampex 10 software. Whole-cell capacitance was calculated using WinWCP V4.7 software from the area under the membrane discharge curve from a clean averaged set of −10 mV hyperpolarizing pulses before applying compensation. Current density was calculated by dividing the peak amplitude by the whole-cell capacitance.

**Patch-clamp recordings of whole-cell mIPSCs.** Whole-cell mIPSCs were recorded in hippocampal neuron cultures at 14DIV after transfection with either shRNA#1 or scrambled shRNA#1 at 8DIV. mIPSCs were recorded at a holding potential of −70 mV over 2-5 min, filtered at 2 kHz and digitized at 20 kHz using Clampex 10.1 software. Neurons were perfused with external solution (mM: 130 NaCl, 2.5 KCl, 2.2 CaCl$_2$, 1.5 MgCl$_2$, 10 D-glucose, 10 HEPES–NaOH, pH 7.4). Sodium channel blocker (500 μM lidocaine) and NMDA, AMPA/kainate receptor blocker (3 mM kynurenic acid) were added to the external solution before recording. The pipette solution was caesium chloride based (140 CsCl, 2 MgCl$_2$, 1 CaCl$_2$, 10 EGTA, 10 HEPES, 2 ATP (disodium salt) mM, adjusted to pH 7.3 with CsOH). Recordings were performed with a Multiclamp 700B amplifier (Axon CNS Molecular Devices, USA). Pipette resistance was 2–3 MΩ and series resistance was always below 20 MΩ. Analyses were performed offline with pCLAMP 10.1 software using a threshold crossing principle; the detection level was set at 5 pA. Raw data were inspected visually to eliminate false events; data from cells with noisy or unstable baselines were also discarded. mIPSC population averages were obtained by aligning events at the mid-point of the rising phase.

**Patch-clamp recordings on acute hippocampal slices.** Adult (250–300 g) male Sprague Dawley CD rats were used. The animals were injected with AAV5-scrambled shRNA#1 (or #2) (left hemisphere) and AAV5-shRNA#1 (or #2) (right hemisphere). Three weeks later the animals were anaesthetized in a chamber saturated with chloroform and then decapitated. The brain was removed rapidly and for patch-clamp recordings placed in ice-cold cutting solution (220 mM sucrose, 2 mM KCl, 1.3 mM NaH$_2$PO$_4$, 12 mM MgSO$_4$, 0.2 mM CaCl$_2$, 10 mM glucose, 3 mM kynurenic acid and 2.6 mM NaHCO$_3$, pH 7.3, perfused with 95% O$_2$/5% CO$_2$),

For recording fEPSPs, the brain was placed in oxygenated (95% O$_2$/5% CO$_2$) ice-cold artificial CSF (aCSF; 125 mM NaCl, 2.5 mM KCl, 1.25 mM NaH$_2$PO$_4$, 1 mM MgCl$_2$, 2 mM CaCl$_2$, 25 mM glucose, and 26 mM NaHCO$_3$, pH 7.3). Thickness of 300–400 μm coronal hippocampal slices were cut with a VT1000 S vibratome (Leica) and incubated 40 min at 36 °C (patch-clamp) or room temperature (fEPSPs) in aCSF. Slices were transferred to a recording chamber perfused with aCSF (~2 ml min$^{-1}$ at 32 °C). Whole-cell patch-clamp and fEPSP recordings were performed with a Multiclamp 700B amplifier (Axon CNS molecular devices, USA) using an infrared-differential interference contrast microscope. Borosilicate glass microelectrodes outer diameter 1.5 μm (Sutter Instruments) were prepared with a four-step horizontal puller (Sutter Instruments) and had a resistance of 3–5 MΩ.

Whole-cell GABA$_B$R-mediated currents where recorded from CA1 pyramidal neurons at a holding potential −80 mV following perfusion of baclofen 100 μM to evoke GABA$_B$R currents and kynurenic acid (3 mM) to block glutamatergic transmission. At the end of each recording, the slices were perfused with GABA$_B$R antagonist CGP55845 (5 μM) to check current specificity. The pipettes were filled with (126 K gluconate, 4 NaCl, 1 EGTA, 1 MgSO$_4$, 0.5 CaCl$_2$, 3 ATP (Mg salt), 0.1 GTP (Na salt), 10 glucose, 10 HEPES–KOH mM, pH 7.3).

fEPSPs were evoked by stimulation (frequency 0.05 Hz) of the Schaffer collateral pathway of the CA1 region using aCSF-filled monopolar glass pipettes. fEPSPs were recorded (acquired at 20 kHz, filtered at 5 kHz) from the dendritic field of CA1 pyramidal neurons again using aCSF-filled electrodes. Input–output (I–O) curves were constructed by measuring the slope of fEPSPs in response to stimulation with increasing intensity (0–1.0 mA). Stimulus strength was adjusted to give 50% maximal response. Long-term potentiation was elicited by high frequency stimulation (100 stimuli at 250 Hz) while long-term depression was elicited by low frequency stimulation (900 stimuli at 1 Hz). Recordings were acquired using Clampex 10.1 software and analysed offline with Clampfit 10.1 software.

For whole-cell patch-clamp electrophysiological recordings on dentate gyrus granule cells, coronal hippocampal slices (thickness, 250–300 μm) were prepared and incubated first for 40 min at 36 °C and then for 30 min at room temperature in oxygenated (95% O$_2$/5% CO$_2$). aCSF. Slices were transferred to a recording chamber perfused with aCSF at 33 °C temperature at a rate of about 2 ml min$^{-1}$.

Tonic GABAergic currents and mIPSCs were recorded at a holding potential of −65 mV in the presence of kynurenic acid (3 mM) with the caesium chloride internal solution (above) supplemented with 5 mM QX-314 (lidocaine N-ethyl bromide, only for tonic currents).

Access resistance was between 10 and 20 MΩ; if it changed by >20% during the recording, the recording was discarded. For tonic currents recordings, after a baseline period of 2–5 min, GABA (5 μM) or THIP (3 μM) was added to the aCSF to increase the tonic component of the GABAergic transmission and to measure modification in extrasynaptic GABA$_A$R subunits composition. At the end of the experiments bicuculline (20 μM) was added to block all GABAergic currents. For the recording of mIPSCs, lidocaine (500 μM) was added in the external solution and recordings were performed as previously described for culture experiments. Analysis was performed offline with Clampfit 10.1 software. CPG 54628, the selective GABA$_B$R antagonist, was used at the final concentration of 5 μM.

**Behavioural tests.** Behavioural tests were carried out 3 weeks after rats were injected bilaterally with either Shrm4 AAV5-shRNA#1 (or #2) or AAV5-scrambled shRNA#1 (or #2). The rats were maintained in 2 per cage post injections before experiments were performed. Behavioural tests were carried out in the following order with a gap of a week between tests: spontaneous motor activity, elevated plus maze, marble-burying, EEG and PTZ-induced seizures. Ten animals per condition were submitted to the behavioural tests for 3 consecutive weeks and during the fourth and fifth weeks, animals of each condition were divided in two subgroups: one submitted to PTZ-induced seizures and the other to EEG. Behavioural experiments were carried out during the light phase of the light/dark cycle between 1000 hours and 1400 hours, and performed by trained observers blind to treatment.

**Spontaneous motor activity test.** Motor measurements were taken in AAV5-shRNA#1 (or #2) or AAV5-scrambled shRNA#1 (or #2) bilaterally injected rats. Spontaneous motor activity was evaluated as previously described[83] in an activity cage with the following dimensions: 43 cm long × 43 cm wide × 32 cm high (Ugo Basile, Varese, Italy), placed in a sound attenuating room. The cage was fitted with two parallel horizontal infrared beams 2 cm off the floor. Cumulative horizontal movements were counted every 10 min for 30.

**Elevated plus maze test.** The elevated plus maze procedure was used as described previously[84] to investigate anxiety-related behaviour in rats. The apparatus consisted of two open arms (50 cm × 10 cm) and two closed arms (50 cm × 10 cm × 40 cm) at right angles, extending from a central platform (10 cm × 10 cm). The apparatus was placed 50 cm above the floor in the centre of a small quiet room in dim light (about 30 lux). Testing was conducted during the early light phase (0930–13.30 hours) of the cycle. After 20 min adaptation to the novel surroundings, an animal was placed on the central platform facing an open arm. The number of open- and closed-arm entries, and time spent in open arms were recorded over 5-min periods.

**Marble-burying test.** This test of anxiety was used according to ref. 85. The test was conducted in a 43 cm × 26 cm × 22 cm box cage with 5 cm of fresh hardwood chip bedding. Each animal was habituated for 15 min to the cage. Then, an array of 24 standard marbles was arranged uniformly over the surface. Individual subjects were placed again in the test cage for 15 min. The number of marbles buried and the latency to the first burying was recorded. A marble was scored as buried if more than two-thirds of it was covered with sawdust. New bedding was used for each animal and marbles were cleaned with 10% acetic acid solution between animals. The same subjects were used for this test and in the elevated plus maze and were tested in a counterbalanced order.

**Sociability and social novelty test.** Social behaviour was carried out in a three-chamber apparatus according to Leite et al.[85] The apparatus was an acrylic rectangular box divided into three compartments of equal size (35 cm height, 50 cm width, 50 cm deep) provided with doors. The sociability test was divided in three sequential phases of 10 min each. During the habituation period, the test rat was placed in the middle chamber for 10 min where the rat was free to explore the three compartments. Each of the two sides contained an identical empty wire cage. In the sociability phase, an unfamiliar rat (stranger 1) of the same strain, sex and weight was enclosed in one of the wire cages and the time spent in each compartment with the object or social stimulus was measured. In the social novelty phase, a new unfamiliar rat (stranger 2) was enclosed into the wire cage in the opposite compartment and the time spent in each compartment was measured. Before the introduction of a social stimulus the test rat was trapped in the central chamber. The test was videotaped and the time spent in each compartment was measured offline.

**Tube test.** The tube test is a well-known testing paradigm designed to measure social hierarchies, and thus is relevant when investigating social dominance in mice[86] but it can be adapted to rats[87]. This test measures dominant/submissive behaviour without allowing them to fight and injure each other. Rats were initially habituated to the testing apparatus, which consisted of a 10 cm (diameter) by 50 cm (length) transparent plastic tube, of sufficient size to allow one but not two rats to move through the tube. Over two consecutive days, rats were allowed to run through the tube on eight occasions, with alternate trials in which the entry and exit ends were switched. Competition trials involved simultaneously releasing two competing rats into opposite ends of the tube. The individual rat that was able to travel forwards through the tube to exit the other side 'won' and was deemed dominant; the rat that retreated was considered subordinate. The number of wins (%) on the total number of competitions was measured.

**Pentylenetetrazole-induced seizures.** Animals (5 per condition) previously injected (CA1) with AAV5-shRNA#1 or scrambled AAV5-shRNA#1, were injected with pentylenetetrazole ($45 \, mg \, kg^{-1}$, i.p.) (Sigma, St. Louis, MO, USA). Seizures was monitored for 30 min after administration and rated on the Racine scale (1972) (score: 0, no change; I, myoclonic jerks; II, minimum seizures, with convulsive wave through body; III, fully-developed minimal seizures, clonus of head muscles and forelimbs, righting reflex present; IV, major seizures (generalized without tonic phase); V, generalized tonic-clonic seizures. The latency (s) to the first seizure and total time spent in seizures (IV and V scale) were recorded.

**Electroencephalogram recording.** AAV5-scrambled#1 and AAV5-shRNA#1 rats were anaesthetized with an i.p. injection of chloral hydrate dissolved in saline and given at a volume of $10 \, ml \, kg^{-1}$. Under anaesthesia ($450 \, mg \, kg^{-1}$, body weight of chloral hydrate, i.p.), all the rats were placed in a stereotaxic instrument and four silver–silver chloride ball electrodes were fixed epidurally with dental acrylic cement, as described in detail elsewhere[88], on the right and left of the parieto-occipital cortex according to the Paxinos and Watson brain atlas, 2 mm anterior, 2 mm lateral from the midline and 3 mm posterior from the bregma. The four electrodes, and a fifth inserted into the nasal bone and used as ground, were connected to a microconnector attached to the rat's head with dental cement (New Galetti e Rossi, Milan, Italy). Animals were treated with ceftriaxone ($50 \, mg \, kg^{-1}$ i.p.) for three days. One week after electrode placement the rats were allowed to acclimatize themselves to a sound-attenuated Faraday chamber for 1 h a day for 3 days. Then, each freely moving awake rat was connected to the microconnector. Signals were amplified by an Animal Bio-Amplifier (AD Instruments), band-pass filtered (0.2–30 Hz) and then connected to a PC for signal acquisition (PowerLab system, AD Instruments, Castle Hill, Australia) at a sampling rate of 100 Hz and a resolution of 0.2 Hz. Each animal was continuously recorded for 24 h, under basal conditions. All collected data were analysed for abnormalities by an experienced observer blinded to rat condition. All EEG traces were scored for the presence of isolated spikes or repetitive spiking using an additional software (LabChart v8 Pro Windows). Spikes were defined as having a duration < 200 ms with baseline amplitude set to 4.5 times the standard deviation of the EEG signal (determined during inter-spike activity periods, whereas repetitive spiking activity was defined as three or more spikes lasting < 5 s).

**Immunohistochemistry.** AAV-injected rats have been anesthetized with zoletil100 (0.1 ml per 100 g) and xylazine (0.1 ml per 100 g) and transcardially perfused with PBS and PFA 4%/ Sucrose 4%. Dissected rat brains have been fixed ON in PFA 4%, rinsed in PBS and cryoprotected with OCT. The blocks have been sectioned (25 μm-thick) with a cryostat. Sections have been collected on glass slides, blocked with 3% BSA,10% goat serum, 0.4% Triton in PBS for 1 h at RT and then incubated with anti-PSD95 antibody (1:300 in blocking solution, Neuromab) at 4 °C ON followed by three washes with PBS. Sections have been incubated for 1 hr with the secondary antibody (1:400, Alexa 555, Life Technlogies) at RT in blocking solution and washed with PBS. Sections have been mounted on coated super-adhesive glass slides and covered with Mowiol prior to confocal imaging. Images have been deconvoluted with ImageJ plugin Iterative Deconvolve 3D.

**Immuno-Electron Microscopy.** Paraformaldehyde fixed neurons were incubated with a polyclonal primary anti-Shrm4 diluted (1:100) in PBS containing 5% normal goat serum, 0.1% saponin for 1 h; they were then incubated with a secondary antibody conjugated with 1.4 nm gold particles (Life Technologies, CA, US) and fixed with 1% glutaraldehyde in PBS. After washing, gold enhancement was performed using the GoldEnhance EM 2113 kit (Nanoprobes, NY, US). Cells were postfixed with 0.2% osmium tetroxide in 0.1 M phosphate buffer, stained with 0.25% uranyl acetate, dehydrated and embedded in epoxy resin. The distribution of gold particles on ultrathin sections of cortical neurons was assessed adapting a randomness test (Mayhew et al.,[89]). To evaluate the distribution of Shrm4 on our sections, the following compartments were defined: synaptic boutons, post-synaptic terminals, dendrites, un-determined structures and areas where cell structure is absent.

**Data availability.** The authors declare that the data supporting the findings of this study are available within the paper and its Supplementary Information files.

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

## Acknowledgements

We thank Don Ward for language assistance. J.Z. and C.H. were supported by Marie Curie Actions; SyMBaD programme. M.P. was supported by Telethon Italy (TDMP11611TC), (GGP12097), (GGP15167A) and Fondation Lejeune. L.P. was supported by Fondazione Umberto Veronesi, Milan. B.B. was supported by the Swiss National Science Foundation (31003A-152970). C.S. was supported by Telethon (GGP11095), Ministry of Health in the frame of ERA-NET NEURON and PNR-CNR Aging Programme 2012–2014. M.P. and C.S. were supported by Fondazione CARIPLO (Project No. 2012-0593). S.H. and T.G.S. were funded by the Medical Research Council U.K., The Rosetrees Trust and the Leverhulme Trust.

## Author contributions

J.Z., E.M. and M.P. conceived the original idea and wrote the manuscript. J.Z. and E.M. designed and performed the majority of the experiments. S.H. and L.M. designed and performed electrophysiological experiments. A.L., C.H. and P.V. designed and perfomed biochemical experiments. D.M. designed and performed dSTORM experiments. L.B. designed and performed electron microscopy experiments. M.F. designed and performed time-lapse experiment. L.P. and D.B. designed and performed behavioural tests. M.S., M.F., J.H., V.K., F.F., C.S., B.B., S.B. and T.G.S. provided useful comments and review the manuscript. M.P. coordinated the entire project and obtained the main source of fundings.

## Additional information

**Competing financial interests:** The authors declare no competing financial interests.

