## [Peer Review File · Nature Communications]

Reviewers' comments:

Reviewer #1 (Remarks to the Author):

Zapata et al. study the epilepsy and ID-linked gene product Shrm4 in neurons. They report physical and functional interactions with the GABA-B receptor system. Using shRNA-mediated knockdown in cells and in brain, they investigate functional consequences of reduced Shrm4 gene expression, using cell biological, electrophysiological and behavioral experiments.

The study is well done and contains a number of complementary approaches to support the message. It is in general in the scope of Nature Communications.

However, I have major concerns, which preclude publication in its present form. They should be fully addressed prior to publication.

1. The specificity of the Shrm4 antibody needs to be shown, compared to Shrm1-3. Individual Shrm 1-4 expression in a Shrm-negative cell line should check whether Shrm4 antibody detects other family members in western and immunocytochemistry.
2. Page 7, line 7 concludes that Shrm4 is crucial for synaptogenesis. This should be downplayed. Alternatively the effects may be due to a role in synapse maintenance.
3. Page 8, line 207 claims a direct interaction between Shrm4 and DIC. This conclusion is not justified by the IP. It may well be that other factors, such as light chains or accessory factors mediate this interaction.
4. Since microtubules display a mixed polarity in dendrites, dynein may indeed be an anterograde motor for GABABRs, although dynein often mediates removal of many surface membrane proteins towards lysosomes. As a control, the Shrm4 immunoprecipitation exp. should be detected with a pan-KIF5 (kinesin-1) antibody.
5. It is surprising that p150cc1 expression does not affect axonal trafficking of presynaptic GABABRs, since axons show a uniform microtubule orientation (plus ends out) and dynein may be the retrograde GABABR transporter in axons. Did the authors identify an increase of presynaptic receptors upon dynein blockade? Live cell GABABR surface staining with a presynaptic marker could clarify this. In general, the consequences of dynein blockade should be studied in more detail with respect to axons and dendrites.
6. Figure 1c, h; Figure 2b, Figure 3g, Figure 4a, b: full neurons should be shown in the supplements to show GABABRs in the neuronal somata that contain Golgi and lysosomes.
7. The FRAP experiment is difficult to interpret, as it does not distinguish between transport and diffusion. GABABRs represent surface receptors that also undergo lateral plasma membrane diffusion. Shrm4-GABABR co-transport needs to be confirmed in a neuronal time-lapse microscopy assay.
8. The time points of the spine analysis and LTP analysis do not fit together. Could the LTP experiment be performed at an earlier timepoint? Is LTD altered under the spine phenotype observed?
9. GABABRs on spines are functionally implicated with NMDAR-mediated functions. Does Shrm4 knockdown affect excitatory transmission?
10. The in vivo data shown in Figure 6 associate Shrm4 with behavior and seizure propensity. This link has been previously discussed in the literature (references 2 and 5). They do not link Shrm4-

GABABR interactions or GABABR trafficking to seizure propensity. Since this is the message of the study, causal evidence is required. Competitive overexpression of GABABR-Shrm4 interaction motifs (either the receptor tail or Shrm4 PDZ domain) or alternative approaches may be suitable to provide the *in vivo* link.

Reviewer #2 (Remarks to the Author):

Shrm4 is an actin-binding protein implicated in cognitive functioning of the brain and its mutations are known to be associated with human X-linked intellectual disability (XLID). In this manuscript, the authors successfully clarify the functions of Shrm4 in neuronal cells; specifically, they find that Shrm4 interconnects GABAB receptors (GABABRs) and dynein to facilitate the dendritic surface expression of GABABRs. The authors demonstrate that Shrm4 regulates spine morphology through GABABR-Ca²⁺ signaling and links GABABR and the extrasynaptic δ subunit-containing GABAARs, which in turn modulates tonic inhibition. They further showed that mice in which Shrm4 has been silenced exhibit increased anxiety and seizure susceptibility, both of which are major comorbidities of XLID patients.

The authors' findings are highly novel, and this paper provides important information for our understanding of the pathogenesis of ID. The authors' experiments are well designed with appropriate controls and rescue experiments. However, I have several criticisms that I believe should be addressed before publication.

1. Figure 1g is not convincing. To improve the reliability of the data, dose dependency of the inhibitory effect of the Tat 859-870 peptide should be examined.
2. Shrm4 silencing affects spontaneous dendritic Ca²⁺ signals (Figure 2l-2n). To test that this effect is mediated by GABABR surface expression, the authors should examine whether the changes in Ca²⁺ signaling are rescued by exogenous expression of GABAB1 or a GABAB1 mutant lacking the Shrm4-binding sequence.
3. Figure 3a. Colocalization of GABABRs and giantin does not appear to be significant. Thus, the correlation coefficient values should be shown. The staining pattern of intracellular GABABRs suggests that GABABRs may be stacked in other compartments in the secretory pathway. The authors should perform colocalization analysis using ER-, ERGIC- and TGN-specific markers, as well as other cis- and medial-Golgi markers. Does the organelle-specific accumulation of GABABRs occur in the dendrites of Shrm4-silenced neurons (dendritic ER, Golgi outposts, and other) ?
4. Is there a significant difference in seizure severity between AAV5-shRNA#1- and AAV5-scrambled#1-injected rats?
5. The authors should examine adaptive and cognitive behaviors (social behavior, social memory, aggression, etc.), which are frequently compromised in XLID patients.
6. It has recently been reported that impairment in dynein-mediated GABAAR surface expression in neurons is closely associated with autism. The authors should discuss the roles of GABA receptor transport in neurodevelopmental disorders.
7. The authors analyzed GABABR transport to the dendritic surface in Figure 1 and performed electrophysiological analysis related to surface-expressed GABABRs in Figure 4c-4e and Figure 5. However, they analyzed GABABR dendritic transport (soma to dendrite and/or intradendritic transport) in Figure 4a and 4b. I think that the authors should examine surface-expressed GABABRs by surface labeling or surface biotinylation in Figure 4a and 4b.
8. The manuscript contains an overwhelming amount of data and is rather difficult to understand. I feel the manuscript needs to be carefully revised.

Minor points:

1. The title of Supplementary Figure 1 is not appropriate. Immunofluorescent staining and electron

microscopy showed that Shrm4 is also localized presynaptically.

2. The legend of Supplementary Figure 1, line 15: "S" should be replaced with "SYN"
3. Figure 1b. The authors should indicate in the panel which band corresponds to GABAB1a and GABAB1b, respectively.
4. Supplementary Figure 2b. To avoid confusion, the authors should clearly state that the polyclonal anti-Shrm4 antibody used in this study recognizes rat Shrm4, but not human Shrm4.
5. The authors confirmed the isoform specificity of anti-GABAB1 antibody in Supplementary Figure 3a. However, the antibody was already used in Figure 1b, 1c, 1d, 1f and 1g.
6. Figure 1h. The authors should clearly state that the anti-GABAB1 antibody used in surface labeling experiments recognizes the extracellular region of GABAB1.
7. Figures should be numbered in the order of their appearance in the text (e.g. Supplementary Figure 2, Supplementary Figure 6, etc.).
8. The title of Figure 1 should be revised.
9. Figure 2. Although Figure 2n is cited in the text, panel n is not labeled in the figure.
10. Supplementary Figure 5h. FRAP data provides clear visual evidence for the involvement of Shrm4 and dynein in GABABR transport. This figure should be transferred to the main figure. "R1-RFP" (in a -5s image) → "B1-RFP" (GABAB1-RFP)?
11. Figure 4d is not cited in the text.

Reviewer #3 (Remarks to the Author):

Zapata, Moretto et al. describe the new interaction between Shrm4, intellectual disability related protein, protein and GABA_B receptors and role of Shrm4 (and likely its interaction with GABA_B receptors) in several aspects of neuron cell biology as well as in behavior and susceptibility to seizures. First, authors using biochemical assays showed that the interaction is direct and likely needed for surface appearance of GABA_B receptors, proper composition of postsynaptic terminals and dendritic spine morphology. Additionally they pointed to a role of Shrm4 in intracellular trafficking of the receptor along the microtubules. Subsequently they describe role of Shrm4 and likely its interaction with GABA_B receptors in control of inhibitory transmission in vitro and in vivo. Finally they showed impact of Shrm4 knockdown on animal anxiety and seizures development. This is very comprehensive work, which addresses scientific problem on several levels of complexity and provides background information needed to understand molecular and cellular mechanisms underlying mental disability and epilepsy, related to mutations in this gene found in humans. I have no major concerns about this work and I believe it is novel, well done (with adequate methods) study which will be of interest to neuroscientific community, some cell biologists and clinicians. In my opinion no obligatory experiments should be added but I would like to suggest some minor additions if possible:

1. In Fig. 1h data for rescue experiments should be provided
2. It would greatly strength conclusions of this work if authors could provide direct functional evidence for causative relationship between lack of Shrm4, dynein-based trafficking of GABA_B and for example spine or synaptic deficits. One potential way of doing such experiment would be to artificially enforce interaction of GABA_B with dynein in cells lacking Shrm4, using light induced dimerization or chemical induced dimerization.

Reviewer #1

Zapata et al. study the epilepsy and ID-linked gene product Shrm4 in neurons. They report physical and functional interactions with the GABA-B receptor system. Using shRNA-mediated knockdown in cells and in brain, they investigate functional consequences of reduced Shrm4 gene expression, using cell biological, electrophysiological and behavioral experiments.

The study is well done and contains a number of complementary approaches to support the message. It is in general in the scope of Nature Communications.

However, I have major concerns, which preclude publication in its present form. They should be fully addressed prior to publication.

We thank the reviewer for his assessment and are encouraged that the work is perceived as in the scope of Nature Communications. We have, as advised, now performed additional experiments and these, we believe, have improved the quality of our study.

1. The specificity of the Shrm4 antibody needs to be shown, compared to Shrm1-3. Individual Shrm 1-4 expression in a Shrm-negative cell line should check whether Shrm4 antibody detects other family members in western and immunochemistry.

This is a fair point – the antibody we have used is specific (produced by Jeffrey Hildebrand) and has been validated previously (Yoder, M. & Hildebrand, J.D. 2007, Cell motility and the cytoskeleton 64, 49-63). This has now been discussed in the manuscript.

2. Page 7, line 7 concludes that Shrm4 is crucial for synaptogenesis. This should be downplayed. Alternatively the effects may be due to a role in synapse maintenance.

This has now been changed, as suggested, in the revised version of the manuscript.

3. Page 8, line 207 claims a direct interaction between Shrm4 and DIC. This conclusion is not justified by the IP. It may well be that other factors, such as light chains or accessory factors mediate this interaction.

A good point. As suggested, this information has been corrected in the revised version of the manuscript and we moved the assertion that Shrm4 and DIC can directly interact to later on in the manuscript after the in vitro pull down experiments (Fig. 3e of the revised manuscript). Of course, it is still possible that other factors are involved in the interaction observed in brain lysates and this has also been discussed in the manuscript.

4. Since microtubules display a mixed polarity in dendrites, dynein may indeed be an anterograde motor for GABABRs, although dynein often mediates removal of many surface membrane proteins towards lysosomes. As a control, the Shrm4 immunoprecipitation exp. should be detected with a pan-KIF5 (kinesin-1) antibody.

We have now performed co-immunoprecipitation experiments using anti-KIF5B and KIF5C (Niclas et al., 1994, Neuron 12: 1059-1072) antibodies for western-blot following anti-Shrm4 immunoprecipitation.

KIF5B and KIF5C, similar to KIF5A, are not associated with Shrm4. These additional controls are now included in supplementary Fig. 6f of the revised manuscript.

5. It is surprising that p150cc1 expression does not affect axonal trafficking of presynaptic GABABRs, since axons show a uniform microtubule orientation (plus ends out) and dynein may be the retrograde GABABR transporter in axons. Did the authors identify an increase of presynaptic receptors upon dynein blockade? Live cell GABABR surface staining with a presynaptic marker could clarify this. In general, the consequences of dynein blockade should be studied in more detail with respect to axons and dendrites.

Indeed, we did not detect any difference in endogenous GABA_BRs following p150-cc1 expression in axons (Fig. 4a, b). This suggests that dynein is specifically involved in the dendritic transport of GABA_BRs. In addition, we found that neurons transfected at DIV7 with p150-cc1 showed lower peak K⁺ current densities (pA/pF) induced by the GABA_BR agonist baclofen (10 – 100 μM) at DIV14 (Fig. 4d). In this regard, Kinesin-1 has been reported to be a key determinant for GABA_BR axonal localization (Valdés et al., 2012, PLoS ONE 7(8): e44168). Furthermore, knockout of the predominantly presynaptic isoform of GABA_BR in mice (GABA_{B1a}^{-/-}) caused impairment in LTP while GABA_{B1b}^{-/-} mice showed normal LTP (Vigot et al., 2006, Neuron, 18;50(4):589-601.). Hence, it is likely that axonal transport is not affected in our case as normal LTP has been observed in Shrm4 depleted mice.

6. Figure 1c, h; Figure 2b, Figure 3g, Figure 4a, b: full neurons should be shown in the supplements to show GABABRs in the neuronal somata that contain Golgi and lysosomes.

These have now been changed in the revised manuscript (Respectively : Fig. 1c-> Supplementary Fig. 1e ; Fig. 1h-> Supplementary Fig. 2c ; Fig. 2b->Supplementary Fig. 3 ; Fig. 3g contained already the full field of view ; Fig. 4a->Supplementary Fig. 7a ; Fig. 4b-> Supplementary Fig. 7b)

7. The FRAP experiment is difficult to interpret, as it does not distinguish between transport and diffusion. GABABRs represent surface receptors that also undergo lateral plasma membrane diffusion. Shrm4-GABABR co-transport needs to be confirmed in a neuronal time-lapse microscopy assay.

As suggested by the reviewer, we have attempted to perform a time-lapse microscopy assay in neurons to analyse Shrm4-GABA_BRs co-transport. However, we were unable to follow both proteins in our neurons in a timelapse assay possibly because of the large size of the Shrm4 protein (around 200 kDa) which makes recombinant expression problematic with likely toxic effects from coexpression together with GABA_{B1} and GABA_{B2}. We agree with the reviewer that a FRAP experiment cannot distinguish between the two transport mechanisms and this aspect has now been discussed in the manuscript.

8. The time points of the spine analysis and LTP analysis do not fit together. Could the LTP experiment be performed at an earlier timepoint? Is LTD altered under the spine phenotype observed?

We thank the reviewer for raising this point which is certainly of great interest. Unfortunately we do not possess the expertise to perform our experiments with AAV5 infections in juvenile rats and it was thus impossible to perform LTP at earlier timepoints within the time constraints of the revision deadline. Nevertheless, we feel that although these experiments will be of interest, the data will not add to the main conclusions of the manuscript and will be beyond the main scope of this work on Shrm4's role on GABA_BRs transport and cell membrane function.

As suggested by the reviewer, we have now performed LTD experiments on Shrm4-silenced hippocampal slices and these show no significant change in LTD induction compared to controls (Supplementary Fig. 10c). These results are now included in the manuscript.

9. GABABRs on spines are functionally implicated with NMDAR-mediated functions. Does Shrm4 knockdown affect excitatory transmission?

We have performed an input-output curve to measure basal excitatory synaptic transmission (Supplementary Fig. 10a). Even though we observed a slight shift in the I/O curve of Shrm4-knockdown infected rats toward an increase in basal excitation, we did not find a significant difference compared to control rats suggesting that basal excitatory transmission is unaffected in our model. These results are now discussed in the manuscript.

10. The in vivo data shown in Figure 6 associate Shrm4 with behavior and seizure propensity. This link has been previously discussed in the literature (references 2 and 5). They do not link Shrm4-GABABR interactions or GABABR trafficking to seizure propensity. Since this is the message of the study, causal evidence is required. Competitive overexpression of GABABR-Shrm4 interaction motifs (either the receptor tail or Shrm4 PDZ domain) or alternative approaches may be suitable to provide the in vivo link.

We have performed seizure propensity experiments (Fig 7e,f) following PTZ administration on rats intraperitoneally injected with the cell permeable peptide fragment that mimics the minimal region of GABAB1 required for binding to Shrm4. We show that treatment with this peptide significantly reduces the latency to the first seizure and increases the time spent in seizure compared to animals injected with a control peptide. These findings suggest that the specific disruption of the Shrm4/GABA_{B1} binding is sufficient to increase the propensity towards seizure and thereby validate our model.

Reviewer #2

Shrm4 is an actin-binding protein implicated in cognitive functioning of the brain and its mutations are known to be associated with human X-linked intellectual disability (XLID). In this manuscript, the authors successfully clarify the functions of Shrm4 in neuronal cells; specifically, they find that Shrm4 interconnects GABAB receptors (GABABRs) and dynein to facilitate the dendritic surface expression of GABABRs. The authors demonstrate that Shrm4 regulates spine morphology through GABABR-Ca²⁺ signaling and links GABABR and the extrasynaptic δ subunit-containing GABAARs, which in turn modulates tonic inhibition. They further showed that mice in which Shrm4 has been silenced exhibit increased anxiety and seizure susceptibility, both of which are major comorbidities of XLID patients. The authors' findings are highly novel, and this paper provides important information for our understanding of the pathogenesis of ID. The authors' experiments are well designed with appropriate controls and rescue experiments. However, I have several criticisms that I believe should be addressed before publication.

We thank the reviewer for the encouraging words on the overall quality and impact of our study.

1. Figure 1g is not convincing. To improve the reliability of the data, dose dependency of the inhibitory effect of the Tat 859-870 peptide should be examined.

This is an interesting point. As suggested, we have now incubated lysates of cells overexpressing GABA_{B1}R-GFP with increasing concentrations (5, 10 and 20 μ M) of peptide in GST pull-down experiments using the GST-Shrm4 PDZ and analyzed its inhibitory effect through GABA_{B1} co-precipitation (supplementary Fig 1f). We found that incubation with the Tat 859-870 at 10 and 20 μ M showed a dose-dependent reduction in Shrm4/GABA_{B1} binding compared to the Tat control peptide.

2. Shrm4 silencing affects spontaneous dendritic Ca²⁺ signals (Figure 2i-2n). To test that this effect is mediated by GABABR surface expression, the authors should examine whether the changes in Ca²⁺ signaling are rescued by exogenous expression of GABA_{B1} or a GABA_{B1} mutant lacking the Shrm4-binding sequence.

As suggested by the reviewer, we have now analyzed spontaneous dendritic Ca²⁺ - transients in neurons expressing Shrm4-shRNA and a mutant form of GABA_{B1} that lacks the Shrm4-binding sequence at the C-terminus. We previously observed that the overexpression of GABA_{B1a} or GABA_{B1b} was not sufficient to rescue the trafficking to dendrites of GABA_BRs in Shrm4-silenced or p150-cc1-expressing neurons suggesting that overexpressing GABA_B may not be able to rescue the defect in Ca²⁺ signalling. Consistent with this, overexpression of GABA_{B1} Δ C and GABA_{B2} together with Shrm4 knockdown shRNA showed no significant difference in dendritic Ca²⁺ transients compared to shRNA-expressing neurons (supplementary Fig 5b). Taken together, these results confirm that the Shrm4-GABA_BRs binding is required for cellular Ca²⁺ signalling.

3. Figure 3a. Colocalization of GABABRs and giantin does not appear to be significant. Thus, the correlation coefficient values should be shown. The staining pattern of intracellular GABABRs suggests that GABABRs may be stacked in other compartments in the secretory pathway. The authors should perform colocalization analysis using ER-, ERGIC- and TGN-specific markers, as well as other cis- and medial-Golgi markers. Does the organelle-specific accumulation of GABABRs occur in the dendrites of Shrm4-silenced neurons (dendritic ER, Golgi outposts, and other) ?

We thank the reviewer for this observation and since this is technically correct we have now excluded the colocalization with Giantin to avoid confusion. Indeed, we were able to observe a general increase in the soma concentration (Fig. 3a) of GABA_BRs but we cannot exclude the possibility of trafficking impairments

in compartments other than the Golgi. However, because the message of our study concerns Shrm4 function on GABA_BRs trafficking along dendrites, we think it is not crucial to dissect the specific starting compartment of this intracellular trafficking. In addition, we observed a diffuse labelling of the remaining GABA_BRs in dendrites of Shrm4-silenced neurons, as observed in control neurons.

4. Is there a significant difference in seizure severity between AAV5-shRNA#1- and AAV5-scrambled#1-injected rats?

We have now analyzed the number of tonic-clonic seizures between AAV5-shRNA#1 and scrambled#1-injected rats. AAV5-knockdown-shRNA-injected rats exhibited significantly more tonic-clonic seizures (Racine scale 5) compared to controls injected with AAV5-scrambled#1. These results are now included in new Fig. 7d of the revised manuscript.

5. The authors should examine adaptive and cognitive behaviors (social behavior, social memory, aggression, etc.), which are frequently compromised in XLID patients.

As suggested by the reviewer, we have now performed sociability, social novelty as well as dominant/submissive tests. Shrm4 AAV5-knockdown rats displayed greater aggression in terms of percentage of wins, and impaired sociability and social novelty behaviours (Fig 6e, f). Altogether, these new findings validate our model that shows social defects commonly found in patients with XLID.

6. It has recently been reported that impairment in dynein-mediated GABA_AR surface expression in neurons is closely associated with autism. The authors should discuss the roles of GABA receptor transport in neurodevelopmental disorders.

This has now been included in the revised manuscript.

7. The authors analyzed GABABR transport to the dendritic surface in Figure 1 and performed electrophysiological analysis related to surface-expressed GABABRs in Figure 4c-4e and Figure 5. However, they analyzed GABABR dendritic transport (soma to dendrite and/or intradendritic transport) in Figure 4a and 4b. I think that the authors should examine surface-expressed GABABRs by surface labeling or surface biotinylation in Figure 4a and 4b.

We have analysed the surface expression of dendritic GABA_BRs in Scrambled, shRNA#1, shRNA#2, p150-cc1 and Rescue expressing neurons. We observed that the removal of Shrm4 or the blockade of dynein was sufficient to induce a reduction of dendritic GABA_BRs surface levels. These were restored under rescue conditions (Fig. 1h and supplementary Fig. 2c). This is consistent with the important role played by Shrm4 in targeting GABA_BRs to dendrites from where they are targeted to the cell surface.

8. The manuscript contains an overwhelming amount of data and is rather difficult to understand. I feel the manuscript needs to be carefully revised

We have now reorganized the revised version of the manuscript.

Minor points:

1. The title of Supplementary Figure 1 is not appropriate. Immunofluorescent staining and electron microscopy showed that Shrm4 is also localized presynaptically.

This has now been changed in the revised version of the manuscript.

2. The legend of Supplementary Figure 1, line 15: "S" should be replaced with "SYN"

Corrected.

3. Figure 1b. The authors should indicate in the panel which band corresponds to GABAB1a and GABAB1b, respectively.

Agreed, now changed.

4. Supplementary Figure 2b. To avoid confusion, the authors should clearly state that the polyclonal anti-Shrm4 antibody used in this study recognizes rat Shrm4, but not human Shrm4.

This has now been changed in the revised online methods.

5. The authors confirmed the isoform specificity of anti-GABAB1 antibody in Supplementary Figure 3a. However, the antibody was already used in Figure 1b, 1c, 1d, 1f and 1g.

Modified as suggested

6. Figure 1h. The authors should clearly state that the anti-GABAB1 antibody used in surface labeling experiments recognizes the extracellular region of GABAB1.

This has now been changed in the revised online methods.

7. Figures should be numbered in the order of their appearance in the text (e.g. Supplementary Figure 2, Supplementary Figure 6, etc.).

Changed in the revised version of the manuscript.

8. The title of Figure 1 should be revised.

Modified in the revised manuscript.

9. Figure 2. Although Figure 2n is cited in the text, panel n is not labeled in the figure.

This has now been modified in the revised version of the manuscript.

10. Supplementary Figure 5h. FRAP data provides clear visual evidence for the involvement of Shrm4 and dynein in GABABR transport. This figure should be transferred to the main figure. "R1-RFP" (in a -5s image) → "B1-RFP" (GABAB1-RFP)?

Agreed, and modified as suggested.

11. Figure 4d is not cited in the text.

This has now been changed in the revised version of the manuscript.

Reviewer #3

Zapata, Moretto et al. describe the new interaction between Shrm4, intellectual disability related protein, protein and GABA_B receptors and role of Shrm4 (and likely its interaction with GABA_B receptors) in several aspects of neuron cell biology as well as in behavior and susceptibility to seizures. First, authors using biochemical assays showed that the interaction is direct and likely needed for surface appearance of GABA_B receptors, proper composition of postsynaptic terminals and dendritic spine morphology. Additionally they pointed to a role of Shrm4 in intracellular trafficking of the receptor along the microtubules. Subsequently they describe role of Shrm4 and likely its interaction with GABA_B receptors in control of inhibitory transmission in vitro and in vivo. Finally they showed impact of Shrm4 knockdown on animal anxiety and seizures development. This is very comprehensive work, which addresses scientific problem on several levels of complexity and provides background information needed to understand molecular and cellular mechanisms underlying mental disability and epilepsy, related to mutations in this gene found in humans. I have no major concerns about this work and I believe it is novel, well done (with adequate methods) study which will be of interest to neuroscientific community, some cell biologists and clinicians. In my opinion no obligatory experiments should be added but I would like to suggest some minor additions if possible:

We would like to thank the reviewer for the appreciation of the quality and novelty of our work.

1. In Fig. 1h data for rescue experiments should be provided

We have now performed rescue experiments on GABA_BRs live surface staining and expression of the rescue construct is able to restore cell surface expression in the presence of the knockdown (Fig 1h) demonstrating the specificity of our knockdown.

2. It would greatly strength conclusions of this work if authors could provide direct functional evidence for causative relationship between lack of Shrm4, dynein-based trafficking of GABA_B and for example spine or synaptic deficits. One potential way of doing such experiment would be to artificially enforce interaction of GABA_B with dynein in cells lacking Shrm4, using light induced dimerization or chemical induced dimerization.

We thank the reviewer for this interesting suggestion. However, the approaches proposed are technically unfeasible in our case since the enforced binding of motor proteins to cargoes has been used in transport assays only for vesicles (such as peroxysomes) and not for single receptors (Kapitein et al., 2010; Van Bergeijk et al., 2015). The obligate heterodimerisation between GABA_{B1} and GABA_{B2} will also add to the complexity of the technique. Moreover, the induction of the binding via chemical or light stimulus could result in the fusion of dynein with GABA_BRs and this may prevent correct targeting to the plasma membrane by exocytosis and cell surface expression.

We have therefore adopted our peptide based approach, that we have already shown in the previous version of the manuscript to interfere with the binding between GABA_BRs and Shrm4. We treated hippocampal cultured neurons between DIV8 and DIV13 when the peak of synaptogenesis is known to occur. At DIV17, when the number of mature dendritic spines was quantified, we found that the disruption of GABA_BRs-Shrm4 interaction during synaptogenesis was sufficient to induce a reduction of spines similar to those observed with Shrm4 knockdown neurons thereby linking the adaptor function of Shrm4 with dendritic spine formation. These results are now included in Fig. 2i of the revised manuscript.

REVIEWERS' COMMENTS:

Reviewer #1 (Remarks to the Author):

The authors have adequately addressed most of my concerns. I can now support publication of this excellent study in Nature Communications.

Reviewer #2 (Remarks to the Author):

The manuscript has been adequately revised and is now acceptable for publication.

Reviewer #3 (Remarks to the Author):

I found revisions satisfactory. I believe that this manuscript should be accepted for publication in Nat. Comm.